# Differential Expression of miRNAs Between Young-Onset and Late-Onset Indian Colorectal Carcinoma Patients

**DOI:** 10.3390/ncrna11010010

**Published:** 2025-02-02

**Authors:** Sumaiya Moiz, Barsha Saha, Varsha Mondal, Debarati Bishnu, Biswajit Das, Bodhisattva Bose, Soumen Das, Nirmalya Banerjee, Amitava Dutta, Krishti Chatterjee, Srikanta Goswami, Soma Mukhopadhyay, Sudarshana Basu

**Affiliations:** 1Department of Molecular Biology, Netaji Subhas Chandra Bose Cancer Research Institute, Kolkata 700094, India; msumaiya10@gmail.com (S.M.); biotech.varsha93@gmail.com (V.M.); debaratibishnu@gmail.com (D.B.); 2Biotechnology Research and Innovation Council-National Institute of Biomedical Genomics (BRIC-NIBMG), Kalyani 741251, India; bs1@nibmg.ac.in (B.S.); sg1@nibmg.ac.in (S.G.); 3Regional Centre for Biotechnology, 3rd Milestone, Faridabad-Gurugram Expressway, Faridabad 121001, India; 4Department of Regenerative and Cancer Cell Biology, Albany Medical College, Albany, NY 12208, USA; 5Department of Histopathology, Netaji Subhas Chandra Bose Cancer Hospital, Kolkata 700094, India; dasbisuda78@gmail.com (B.D.); nirmalyapgi@gmail.com (N.B.); me.amitavadutta@gmail.com (A.D.); ckrishti9@gmail.com (K.C.); 6Department of Surgical Oncology, All India Institute of Medical Sciences (AIIMS), Rishikesh 249203, India; bodhisattvabose123@gmail.com; 7Department of General Surgery, Nil Ratan Sircar Medical College and Hospital, Kolkata 700014, India; 8Department of Surgical Oncology, Netaji Subhas Chandra Bose Cancer Hospital, Kolkata 700094, India; soumendoc.das@gmail.com; 9Department of Histopathology, Narayana Superspeciality Hospital, Kolkata 700099, India; 10Department of Pathology, Neotia Bhagirathi Woman and Child Care Centre, Kolkata 700017, India

**Keywords:** miRNA, early-onset colorectal cancer, hsa-miR-1247-3p, hsa-miR-326, hsa-miR-148a-3p

## Abstract

Reports indicate a worldwide increase in the incidence of Early-Onset Colorectal Carcinoma (EOCRC) (<50 years old). In an effort to understand the different modes of pathogenesis in early-onset CRC, colorectal tumors from EOCRC (<50 years old) and Late-Onset patients (LOCRC; >50 years old) were screened to eliminate microsatellite instability (MSI), nuclear β-catenin, and *APC* mutations, as these are known canonical factors in CRC pathogenesis. Small-RNA sequencing followed by comparative analysis revealed differential expression of 23 miRNAs (microRNAs) specific to EOCRC and 11 miRNAs specific to LOCRC. We validated the top 10 EOCRC DEMs in TCGA-COAD and TCGA-READ cohorts, followed by validation in additional EOCRC and LOCRC cohorts. Our integrated analysis revealed upregulation of hsa-miR-1247-3p and hsa-miR-148a-3p and downregulation of hsa-miR-326 between the two subsets. Experimentally validated targets of the above miRNAs were compared with differentially expressed genes in the TCGA dataset to identify targets with physiological significance in EOCRC development. Our analysis revealed metabolic reprogramming, downregulation of anoikis-regulating pathways, and changes in tissue morphogenesis, potentially leading to anchorage-independent growth and progression of epithelial-mesenchymal transition (EMT). Upregulated targets include proteins present in the basal part of intestinal epithelial cells and genes whose expression is known to correlate with invasion and poor prognosis.

## 1. Introduction

Colorectal cancer (CRC) accounted for 10% of all new cancer cases (males only, all ages) in 2022 and is the third-most prevalent cancer in the world (https://gco.iarc.who.int/media/globocan/factsheets/populations/900-world-fact-sheet.pdf, accessed on 15 July 2024). As per data from Globocan 2022, there were ~43,360 new CRC cases (males only, all ages) in 2022 in India, making it the fourth most prevalent cancer in the country (https://gco.iarc.who.int/media/globocan/factsheets/populations/356-india-fact-sheet.pdf, accessed on 15 July 2024). Projections taking into account aging, population growth, and human development estimate that by 2040, the incidence of CRC in India will increase by more than 60% (both sexes, all ages) (Global Cancer Observatory, Cancer Tomorrow, https://gco.iarc.fr/tomorrow/en, accessed on 15 July 2024). The number of young people (0–49 years) expected to be diagnosed with CRC has been estimated to increase by more than 13% between 2022 and 2030 and by more than 20% between 2022 and 2040 (both sexes) (Global Cancer Observatory, Cancer Tomorrow; https://gco.iarc.fr/tomorrow/en, accessed on 15 July 2024).

Age represents the primary risk factor for CRC [1], although the cumulative risk for early-onset CRC (0–49 years old) in India has increased by 116% in males and 200% in females in the time span of 26 years (1986 to 2012) (Global Cancer Observatory, Cancer Over Time; https://gco.iarc.fr/overtime/, accessed on 15 July 2024). Analysis of CRC incidence via the Surveillance, Epidemiology, and End Results (SEER) data from 2000 to 2019 revealed that adults aged less than 50 years were noted to have an average of 2.4% annual increase in CRC incidence rates, while adults above the age of 65 years had an average of −3.4% change in CRC incidence. Thus, although there is a general decrease in the overall incidence of CRC, the incidence of the disease in young adults (<50 years) has increased worldwide. This has prompted the ACS (American Cancer Society) to revise the standard age for CRC risk screening to 45 years from 50 years [2].

Traditionally CRC has been classified into three molecular subgroups based on the mechanism of carcinogenesis: chromosomal instability (*APC* (adenomatous polyposis coli) inactivation, Wnt (wingless-related integration site) signaling activation, activating *KRAS* (Kirsten rat sarcoma viral oncogene homolog) mutations), defects in DNA mismatch repair (Microsatellite Instability—MSI) and aberrant CpG island hypermethylation and gene silencing (CIMP, *BRAF* (v-RAF murine sarcoma viral oncogene homolog B1) mutations) [3,4]. However, consensus exists that Early-Onset CRC (EOCRC) is pathologically, anatomically, metabolically, and biologically different from Late-Onset CRC (LOCRC) and hence should be investigated and managed differently [5,6,7]. EOCRC tumors were found to be mostly located in the distal colon (80%), particularly the sigmoid colon and the rectum, with a higher prevalence of adverse histological factors such as signet ring cell differentiation, venous invasion, and perineural invasion [8]. The tumors lacked frequent activating *BRAF* or *KRAS* mutations, suggesting that the molecular events in tumor development differed with respect to the late-onset group [8]. Additionally, EOCRC was not frequently associated with precursor adenomatous lesions [8], suggesting that the classical adenoma-to-carcinoma pathway of molecular events [6,9] does not occur in this patient subset.

Recently, gene expression analyses have identified putative targets that indicate altered pathways in EOCRC [10,11,12,13,14]. The MAPK (mitogen-activated protein kinase) pathway appeared to be deregulated in the early-onset sporadic group as compared to PI3K-Akt (phosphatidylinositol 3-kinase/protein kinase B) in the late-onset group [10]. Bioinformatics analysis on microarray data sets to identify EOCRC-linked differentially expressed genes (DEGs) highlighted 108 upregulated genes and 23 downregulated genes [11]. Functional enrichment of the EOCRC-associated upregulated DEGs indicated strong implication of molecular mechanisms involved in vascular smooth muscle contraction signaling pathway [11]. PPI network analysis identified 7 hub genes—*ACTA2* (smooth muscle cell alpha-2 actin), *ACTG2* (actin gamma-2 smooth muscle), *MYH11* (myosin-11), *CALD1* (caldesmon), *MYL9* (myosin regulatory light polypeptide 9), *TPM2* (β-tropomyosin), and *LMOD1* (leiomodin 1) associated with the vascular smooth muscle contraction signaling pathway [11]. Early-onset sporadic tumors lacking canonical genetic aberrations like MSI and Wnt/β-catenin activation were found to be enriched in Ca^2+^/NFAT pathways [12,13]. High-throughput RNA sequencing of EOCRC tumors followed by validation by RT-qPCR identified significant upregulation of genes *TNS1* (tensin 1) and *MET* (MET proto-oncogene, receptor tyrosine kinase/hepatocyte growth factor receptor) [14].

MicroRNAs (miRNAs) are important regulatory molecules that may act as either tumor suppressors or oncogenes depending on the cellular environment in which they are expressed [15]. Analysis of miRNA expression profiles and their predicted target genes can indicate the aberrant physiology of a system and may be targeted in therapy or used as biomarkers for diagnostic purposes [16,17]. miRNAs play important roles in the development of CRC, as their deregulation affects signaling pathways like Wnt/β-catenin, epidermal growth factor receptor, p53, mismatch repair/DNA repair, transforming growth factor beta, PI3K/Akt, and Ras-Raf-MAPK [18,19]. Numerous studies assessing miRNA levels in the blood and tissues of CRC patients have detected their altered expression [18,19,20,21,22,23]. Experimental modulation of wild-type p53 in CRC cell lines was found to upregulate tumor-suppressing miR-34a, miR-192, miR-194, and miR-215 [24,25].

In spite of the bulk of studies investigating miRNAs in CRC, very few have differentially examined miRNAs in EOCRC as compared to LOCRC. Yantiss et al. in 2009 studied the clinical, pathological, and molecular features of young-onset colorectal carcinoma in patients < 40 years old. They observed significant overexpression of miR-21, miR-20a, miR-145, miR-181b, and miR-203 in the tumors of young patients [26]. Investigation of Turkish EOCRC tumors revealed upregulation of miR-106a and downregulation of miR-143 and miR-125b [27]. Elevated expression of miR-106a and downregulation of miR-125b correlated with lymph node metastasis in patients [27]. In this study, EOCRC tumors were compared with normal tissues, and no direct comparison with a LOCRC subset was included in the investigation. Recent work by Nakamura et al. identified a four-miRNA liquid biopsy panel for EOCRC diagnosis that robustly identified patients with EOCRC even in early-stage disease, indicating its clinical effectiveness [28]. RNA-seq of sporadic EOCRC-associated miRNAome and transcriptome and validation by bioinformatics study and RT-qPCR in additional cohorts identified the miR-31-5p-*DMD* axis as a novel biomarker of sporadic EOCRC [29]. *DMD* (dystrophin) was found to be downregulated and miR-31-5p was found to be upregulated in sporadic EOCRC, pointing to its possible role in the occurrence of EOCRC [29]. This study also identified miRNAs significantly altered in LOCRC. They reported elevated levels of miR-31-3p and reduced levels of miR-10b-5p specifically in tumors of late-onset CRC patients as compared to adjacent pericarcinomatous tissue [29]. However, miRNAs deregulated in Indian EOCRC patients have not yet been explored.

The goal of our study was to highlight EOCRC-specific miRNA alterations in Indian cohorts, which could be used to discriminate between EOCRC and LOCRC and potentially identify deregulated molecular pathways in early-onset disease. Highlighting the EOCRC-specificity of the dysregulated miRNAs was important for an insight into the mechanism contributing to the rise in EOCRC cases. To achieve this goal, we performed genome-wide small-RNA sequencing of sporadic colorectal tumors in young patients (<50 years old) and old patients (>50 years old) negative for canonical CRC markers like MSI, nuclear β-catenin, and *APC* mutation. Differentially expressed EOCRC miRNAs (DEMs) were validated by analysis of expression in TCGA-COAD and TCGA-READ datasets followed by quantitative real-time PCR (RT-qPCR) in additional EOCRC and LOCRC patient cohorts. Subsequent bioinformatic analysis of the validated miRNAs identified deregulated pathways in EOCRC. To the best of our knowledge, this study is the first to compare miRNA expression between EOCRC and LOCRC patients in India and additionally identify EOCRC tumor miRNA alterations that are specific to early-onset disease.

## 2. Results

### 2.1. Patient Recruitment and Sample Collection

34 histologically confirmed colorectal adenocarcinomas and adjacent colorectal mucosas were collected. Collected tissue specimens corresponded to 7 young patients (<50 years old) and 27 old patients (>50 years old). Upon histopathological analysis by a trained pathologist, malignant lesions were found to be of the following TNM stages: T1 (11.76%), T2 (29.41%), T3 (41.2%), and T4 (17.65%). 11.76% of them represented well-differentiated adenocarcinoma (Grade: G1), 70.6% were moderately differentiated (Grade: G2), and 14.71% were poorly differentiated (Grade: G3). The screening was conducted to eliminate tumors with MSI, nuclear localization of β-catenin, and *APC* mutations (mutation cluster region—between codons 1260 and 1596 of exon 15 of the APC gene) [30,31,32]. Representative images of immunohistochemical detection of MSI and nuclear β-catenin are shown in Appendix A, respectively.

### 2.2. NGS Small-RNA Sequencing and Analysis

Out of the patients who were microsatellite stable, without nuclear β-catenin and lacking somatic *APC* mutations, tumors and paired normal tissues of 5 (Patient no. 1, 2, 3, 4, and 5) were sent for NGS-based miRNA sequencing (seq). The 5 patients who were chosen for miRNA seq consisted of 3 young patients (mean age: 43 years) and 2 old patients (mean age: 63 years). Pathological information of the clinical samples sent for miRNA-seq is summarized in Table 1. Sequencing data of the ten human samples (normal and tumor tissue of 5 CRC patients) were obtained for the small-RNA (miRNA) transcriptome analysis. A total of twenty paired-end fastq files were used for the small-RNA-seq analysis via a pipeline FastQC-Fastp-SortMeRNA-miRDeep2-edgeR. The schematic workflow for the overall analysis is depicted in Figure 1.

Sequencing the miRNA libraries for CRC tissues and paired normal tissues resulted in a total of 91,658,630 and 94,627,488 raw reads, respectively. The removal of adaptor sequences, junk reads, reads other than 15 to 30 bp, rRNA, snRNA, snoRNA, and tRNA produced 83,719,335 and 90,030,493 clean reads respectively. The summary of read alignment is indicated in Appendix A. At least 85% of the raw reads accounted for the clean reads, which suggested that a useful group of miRNAs was obtained with a reasonable sequencing depth. Overall mapping rate was over 28% among samples. Read depth coverage and sequence length distribution plots are depicted in Appendix A.

### 2.3. Differential miRNA Expression Analysis

For single-sample comparisons, normal colonic mucosa (pericarcinomatous tissue at a minimum distance of 5 cm from visible tumor edges) of each patient was taken as the normal control for differential expression analysis. The dataset for the normal control of each patient was compared with the tumor dataset of that patient. For multiple sample comparisons, the patient datasets were divided into two groups—young and old—based on their age. Patients 1, 2, and 3 were considered as young as their age was <50 years, and patients 4 and 5 were considered as old as their age was >50 years. In the Young Normal vs. Tumor comparison, the normal dataset was compared with the tumor dataset of young patients (normal as the control group and tumor as the test group). Similarly, in the Old Normal vs. Tumor comparison, the normal dataset for old patients was compared with the tumor dataset for the same (normal as control group and tumor as test group). Differentially expressed miRNAs (DEMs) were obtained by filtering the results of DEA (Differential Expression Analysis) using *p*-value < 0.05 and Log_2_ Fold Change value at >2 (upregulation) or <−2 (downregulation). The volcano plots for each sample comparison of each patient are depicted in Figure 2A–E. These were generated having cut-off criteria of Log_2_ Fold Change ≥2 (*p* < 0.05) for upregulated and ≤−2 (*p* < 0.05) for downregulated genes.

All identified miRNAs (both known and novel) of Normal_old (N-O), Tumor_old (T-O), Normal_young (N-Y), Tumor_young (T-Y) were used for the generation of the Venn diagram (Figure 2F). A heatmap of the top 20 miRNAs (Figure 2G) and all miRNAs (Appendix A) identified by RNA Sequencing of our 10 samples and sorted by variance in decreasing order was generated. Known DEMs with a cut-off *p*-value < 0.05 and Log_2_ fold change value at >2 or <−2 were identified in the multiple sample comparisons of Young Tumor vs. Normal and Old Tumor vs. Normal (normal as control group and tumor as the test group) and were analyzed by Venn diagram (https://bioinfogp.cnb.csic.es/tools/venny/, accessed on 2 June 2022) [33]. (Figure 2H). 23 DEMs were identified as specific to the Young Tumor vs. Normal group whereas 11 were identified as unique to the Old Tumor vs. Normal group. 5 DEMs were common to both groups indicating overall differential expression in CRC. The 5 common elements differentially expressed in both subsets were hsa-miR-129-5p, hsa-miR-9-5p, hsa-miR-1-3p, hsa-miR-145-5p and hsa-miR-133a-3p. The full list of DEMs identified in both young and old patients is described in Appendix A. The list of identified DEMs specific to young patient tumors (EOCRC) and old patient tumors (LOCRC) are given in Table 2 and Table 3, respectively.

### 2.4. Identification of DEMS to Be Validated in Additional Cohorts Based on TCGA Data Analysis

Out of the 23 DEMs specific to the EOCRC subset, we chose the top 10 DEMs (on the basis of LogFC) for further validation using TCGA datasets. The top 10 miRNAs differentially expressed in young patient tumors of the discovery cohort in our small-RNA sequencing analysis were hsa-miR-1247-3p, hsa-miR-27a-5p, hsa-miR-96-5p, hsa-miR-326, hsa-miR-378c, has-miR-378d, hsa-miR-378a-5p, hsa-miR-378e, hsa-miR148a-3p and hsa-miR-135b-5p (Table 2). These miRNAs were tested on the TCGA datasets of colon adenocarcinoma (COAD) and rectal adenocarcinoma (READ) using matched TCGA normal datasets (CancerMIRNome (jialab-ucr.org), COAD—accessed on 12 June 2022, READ—accessed on 25 December 2024) [34]. We observed significant and constant differential expression of hsa-miR-1247-3p (only in COAD), hsa-miR-27a-5p, hsa-miR-96-5p, hsa-miR-326, hsa-miR-378a-5p, hsa-miR148a-3p and hsa-miR-135b-5p among large data sets (Figure 3A–G). The upregulation or downregulation observed was similar to that obtained by our NGS analysis. Hsa-miR-378e, hsa-miR-378c, and hsa-miR-378d did not show significant differential expression in the TCGA datasets of tumors and matched normals (Figure 3H–J). Hsa-miR-1247-3p was significantly differentially expressed in the COAD dataset but did not show significant expression in the READ dataset. However, it is to be noted that the datasets depicted in Figure 3 represent all ages and do not incorporate any age-specific comparisons. EOCRC-specific differential expression of miRNAs may not be reflected in the overall COAD and READ dataset analysis, as the result would be skewed towards the larger number of LOCRC cases. Since the aim of our study was to understand miRNA differences between EOCRC and LOCRC, it was necessary to analyze age-specific differences in miRNA expression between tumors and adjacent normal tissues.

To visualize age-specific miRNA expression in the TCGA-COAD and TCGA-READ datasets, we divided the population into four subsets—Young Normal, Old Normal, Young Tumor, and Old Tumor. The COAD and READ datasets were integrated to determine the Average CPM (AveCPM) of each subset. miRNA levels (log_2_ AveCPM) were compared between these four subsets and represented graphically (Figure 4).

Hsa-miR-378e did not show any expression in the young dataset (Figure 4H). miRNA levels of hsa-miR-326 were slightly upregulated in young normal as compared to old normal (Figure 4F). Similarly, hsa-miR-378d was slightly downregulated in normal tissues of young patients as compared to those of old patients (Figure 4J). For all other miRNAs, basal miRNA levels in normal tissues for both age groups were similar or close to each other (Figure 4A–E,G,I). Except for hsa-miR-378e, all miRNAs were upregulated in the young tumor dataset population with respect to their normal counterpart (Figure 4A–G,I,J). This is contrary to our previous RNA-seq results, where we observed downregulation for hsa-miR-326, hsa-miR-378a-5p, hsa-miR-378c, and hsa-miR-378d (Table 2). Hsa-miR-135b-5p (Figure 4A) and hsa-miR-378d (Figure 4J) showed upregulation in both young tumor and old tumor subsets (with respect to young normal and old normal, respectively) (Figure 4A,J). Slight upregulation in the old tumor as compared to old normal was observed for hsa-miR-148a-3p (Figure 4B), hsa-miR-27a-5p (Figure 4D), and hsa-miR-378c (Figure 4I). Increase in levels of hsa-miR-1247-3p (Figure 4C), hsa-miR-96-5p (Figure 4E), hsa-miR-326 (Figure 4F), and hsa-miR-378a-5p (Figure 4G) in tumors with respect to normal was observed only in the young datasets (Figure 4C,E–G).

Hsa-miR-326 and hsa-miR-378a-5p are significantly downregulated in TCGA-COAD and TCGA-READ dataset analysis (Figure 3F,G), but age-specific analysis reveals upregulation in young tumor subsets, as compared to corresponding normal (Figure 4F,G). This justifies our argument that age-specific miRNA profiles may differ from overall TCGA data, which tends to reflect the majority LOCRC pattern of miRNA expression.

### 2.5. Validation of Selected DEMs in EOCRC and LOCRC

Validation in the TCGA-COAD and TCGA-READ datasets identified 9 miRNAs that showed differential expression in young tumors as compared to corresponding normal (hsa-miR-1247-3p, hsa-miR-27a-5p, hsa-miR-96-5p, hsa-miR-326, hsa-miR-378a-5p, hsa-miR-148a-3p, hsa-miR-135b-5p, hsa-miR-378c, and hsa-miR-378d). The next step was to validate by RT-qPCR in additional EOCRC (<50 years old) and LOCRC (>50 years old) validation cohorts of 16 young patients and 11 aged patients, respectively (Table 4). Validation in LOCRC cohorts was necessary to verify if the differential expression of miRNAs was pan-cancer or restricted specifically to the young population. Since we were looking for EOCRC-specific miRNAs, it was important that differential expression was restricted to the EOCRC subsets. Adjacent normal colonic mucosa of each tumor was used as the corresponding control. Table 4 details the clinicopathological details of patients, both EOCRC and LOCRC (tumor tissue and normal mucosa), included in the validation cohorts. miRNA levels were detected by RT-qPCR using stem-loop primers for cDNA synthesis, and PCR amplification was performed using specific forward and reverse primers [35,36].

Expression patterns for hsa-miR-135b-5p (Figure 5A), hsa-miR-148a-3p (Figure 5B), hsa-miR-1247-3p (Figure 5C), hsa-miR-27a-5p (Figure 5D), hsa-miR-96-5p (Figure 5E), and hsa-miR-326 (Figure 5F) in all 16 young patients of the EOCRC validation cohorts supported our NGS data analysis results (Figure 5A–F and Table 2). Hsa-miR-135b-5p (Figure 5A), hsa-miR-27a-5p (Figure 5D), and hsa-miR-96-5p (Figure 5E) were significantly upregulated in both EOCRC and LOCRC tumors as compared to respective adjacent normal tissues (Figure 5A,D,E). Thus, these miRNAs do not show age-specific expression. Hsa-miR-148a-3p (Figure 5B) was significantly upregulated specifically in EOCRC tumors with respect to control. The age-wise distribution of hsa-miR-135b-5p (Figure 5A), hsa-miR-148a-3p (Figure 5B), hsa-miR-27a-5p (Figure 5D), and hsa-miR-378a-5p (Figure 5G) observed in our EOCRC and LOCRC cohorts resembled that of age-specific TCGA datasets (Figure 4A,B,D,G). Hsa-miR-96-5p (Figure 5E) showed significant upregulation in both EOCRC and LOCRC cohorts in contrast to EOCRC-specific upregulation observed in TCGA age-specific datasets (Figure 4E). Hsa-miR-378c and hsa-miR-378d were upregulated in old tumors of age-specific TCGA datasets (Figure 4I,J); however, our RT-qPCR validation revealed downregulation in aged tumors as compared to corresponding normal (Figure 5H,I). Differential expression of hsa-miR-1247-3p (Figure 5C) and hsa-miR-326 (Figure 5F) was observed only in EOCRC tumors, with hsa-miR-1247-3p being upregulated and hsa-miR-326 being downregulated with respect to control. Hsa-miR-326 was upregulated in the TCGA young datasets (Figure 4F); however, it was significantly downregulated in young tumors of both our discovery (Table 2, NGS data analysis) and validation EOCRC cohorts (Figure 5F). These findings suggest that hsa-miR-1247-3p and hsa-miR-326 are differentially expressed specifically in EOCRC. Understanding their regulatory pathways may indicate potential therapeutic avenues to be targeted for early disease management.

In contrast to our sequencing data analysis results (Table 2), hsa-miR-378c (Figure 5H) and hsa-miR-378d (Figure 5I) show significant upregulation among young tumor tissues with respect to their adjacent normal colonic mucosa (Figure 5H,I). The upregulation of hsa-miR-378c and hsa-miR-378d in the EOCRC cohort (Figure 5H,I) is similar to that observed in the TCGA age-specific analysis (Figure 4I,J). Significant downregulation of hsa-miR-378a-5p in EOCRC tumors observed in our NGS data analysis (Table 2) was not observed in the validation cohort (Figure 5G).

### 2.6. Comparison of Selected DEMs Between Young and Old Tissues

Validation of the selected DEMs in EOCRC and LOCRC validation cohorts was computed on the basis of their expression in tumor tissues as compared to corresponding adjacent normal tissues (Figure 5). Since we are comparing young and old tumors, it was necessary to verify that basal miRNA levels in adjacent normal tissues used as controls were similar for both EOCRC and LOCRC. The observed difference in expression between EOCRC and LOCRC (Figure 5) could be age-related and hence not specific to early-onset disease. Therefore, to answer this question, we compared normalized Ct values (∆Ct = Ct_miRNA_–Ct_U6_) for the selected DEMs between EOCRC and LOCRC adjacent normal colonic tissues and EOCRC and LOCRC tumor tissues. Ideally, miRNAs involved in early-disease onset should show age-independent expression in normal tissues with significant upregulation/downregulation between young and old tumors.

miRNA levels in adjacent normal tissues of hsa-miR-27a-5p (Figure 6D), hsa-miR-96-5p (Figure 6E), hsa-miR-378c (Figure 6H), and hsa-miR-378d (Figure 6I) differed significantly between young and old patients, suggesting basal age-related differences (Figure 6D,E,H,I, left panel). Comparison between normalized Ct values of hsa-miR-135b-5p (Figure 6A) and hsa-miR-378a-5p (Figure 6G) in adjacent normal and tumor tissues of young and old patients revealed non-significant differences suggesting a lack of EOCRC specificity. The level of upregulation of hsa-miR-148a-3p (Figure 6B) and hsa-miR-1247-3p (Figure 6C) in young tumors was observed to be significantly more than in old tumors (Figure 6B,C, right panel). Hsa-miR-326 was significantly downregulated in young tumors as compared to aged tumors (Figure 6F, right panel), consistent with our previous observations (Figure 5F and Table 2). However, in contrast to age-specific TCGA analysis (Figure 4F), we observed no significant change between young and old adjacent normal tissues (Figure 6F, left panel) and no upregulation in young tumors as compared to aged ones (Figure 6F, right panel). We, therefore, selected hsa-miR-148a-3p, hsa-miR-1247-3p, and hsa-miR-326 as being differentially expressed between young and old patients with definitive roles specifically in the development of EOCRC.

### 2.7. Analysis of DEM Targets Differentially Expressed in TCGA-COAD and TCGA-READ Datasets

Experimentally validated targets of hsa-miR-148a-3p, hsa-miR-1247-3p, and hsa-miR-326 were identified through miRNet (https://www.mirnet.ca/, accessed on 3 July 2024) [37]. Differentially expressed (DE) genes in the TCGA datasets of colon adenocarcinoma (COAD) and rectum adenocarcinoma (READ) with |Log_2_FC| cutoff of 1.00 and a q-value cutoff of 0.01 were identified using GEPIA (http://gepia.cancer-pku.cn/, accessed on 4 July 2024 for COAD accessed on 26 December 2024 for READ) [38]. Target genes of the selected miRNAs were compared by Venn-diagram analysis (https://bioinfogp.cnb.csic.es/tools/venny/, accessed on 26 December 2024) [33] with this DE-gene list to identify DE-target genes altered in colorectal tumor tissue in a direction reciprocal to that of the corresponding miRNA.

Figure 7B and Figure 7C depict the overlap between TCGA-COAD and TCGA-READ downregulated genes and targets of upregulated hsa-miR-1247-3p and hsa-miR-148a-3p, respectively. For miR-1247-3p, we identified 19 miRNA target genes downregulated in both COAD and READ and 1 target gene downregulated only in READ. For miR-148a-3p, 20 miRNA target genes were common to both COAD and READ, 2 target genes were common with only COAD, and 4 targets were common to only READ. In the case of miR-326, 24 miRNA targets were found to overlap with DE-upregulated genes common to both COAD and READ, and 8 targets overlapped with DE-upregulated genes of only READ. Details of predicted DE-miRNA targets for all three miRNAs are given in Appendix A.

We performed a differential gene expression analysis of these COAD and READ downregulated and upregulated targets in normal and tumor colon and rectum adenocarcinoma datasets on TNMplot (https://tnmplot.com/analysis/, accessed on 26 December 2024) [39] (Figure 7E–G). Selected targets were downregulated (targets of hsa-miR-1247-3p and hsa-miR-148a-3p: Figure 7E,F, respectively) and upregulated (targets of hsa-miR-326: Figure 7G) in tumor tissues as compared to normal, indicating their role in oncogenesis and early disease onset.

### 2.8. Pathway Enrichment of TCGA-COAD and TCGA-READ Targets of Validated DEMs in EOCRC

Gene Ontology and Pathway Enrichment Analysis were performed for the genes common between TCGA datasets and targets of our validated differentially expressed miRNAs to identify pathways deregulated in EOCRC. Enrichment analysis was performed using the tool ShinyGO (v.0.81 [40] (http://bioinformatics.sdstate.edu/go/, accessed on 27 December 2024) based on the annotations Biological Process (BP), Molecular Function (MF), and Cellular Component (CC) against humans as a selected species. Enriched upregulated (for miR-326) and downregulated (for miR-148a-3p and miR-1247-3p) genes identified through GO pathway analysis are given in Appendix A.

The validated DEM targets for upregulated hsa-miR-1247-3p analyzed for BP and MF were mostly enriched in processes of ‘anatomical structure morphogenesis’ and ‘receptor-mediated endocytosis’ (Figure 8A) with a molecular function of ‘phosphatidylinositol-3,5-bisphosphate binding’ (Figure 8B). No enrichment was observed for the annotation CC. Validated DEM targets for upregulated hsa-miR-148a-3p were enriched in the biological processes of ‘negative regulation of metabolic processes’, ‘regulation of phosphorus metabolic processes’, ‘tissue morphogenesis’, ‘cellular response to nutrient levels’ and ‘negative regulation of anoikis’ (Figure 8C). Cellular component enrichment was observed in the category of ‘Ruffle membrane’, ‘leading edge membrane’, and ‘RISC complex’ (Figure 8D). Molecular function enrichment of hsa-miR-148a-3p targets was observed in the categories of ‘molecular function regulator activity’, ‘chaperone binding’, ‘protein phosphatase binding’, and ‘phosphatidylinositol-3-phosphatase activity’ (Figure 8E). BP analysis of upregulated hsa-miR-326 targets indicated enrichment for the categories of ‘blood vessel development’, ‘angiogenesis’, ‘response to inorganic substance’, ‘Gland development’, and ‘Icosanoid transport’ (Figure 8F). Cellular component analysis revealed upregulation of proteins enriched in the ‘basolateral plasma membrane’ and the ‘basal part of the cell’ (Figure 8G). Molecular function enrichment was observed in the categories of ‘ATPase-coupled inorganic anion transmembrane transporter activity’ and ‘ABC-type glutathione S-conjugate transporter activity’ (Figure 8H).

## 3. Discussion

The last few decades have seen an increase in the incidence of colorectal cancer among individuals less than 50 years old, also referred to as EOCRC. Not much is known about the EOCRC or the reason for its increase in the younger population. We performed an RNA-Seq analysis of sporadic colorectal tumors in young patients (EOCRC < 50 years old) and aged patients (LOCRC > 50 years old) negative for canonical CRC markers like MSI, nuclear β-catenin, and *APC* mutation. 23 miRNAs were differentially expressed specifically in young patients, 11 miRNAs were differentially expressed specific to aged patients, and 5 miRNAs were found to be differentially expressed in both. The 5 miRNAs (hsa-miR-129-5p, hsa-miR-9-5p, hsa-miR-1-3p, hsa-miR-145-5p, and hsa-miR-133a-3p) common to both EOCRC and LOCRC have been previously reported as downregulated in CRC [41,42,43]. For validation of identified EOCRC DEMS, we divided the TCGA-COAD and TCGA-READ datasets into young (<50 years) and old (>50 years) groups and compared the expression of the top 10 EOCRC miRNAs in normal and tumor samples of the age-specific TCGA cohorts. Hsa-miR-1247-3p, hsa-miR-27a-5p, hsa-miR-96-5p, hsa-miR-148a-3p, hsa-miR-326, hsa-miR-378a-5p, hsa-miR-135b-5p, hsa-miR-378c, and hsa-miR-378d showed differential expression in young tumors as compared to corresponding normal. Interestingly, hsa-miR-326 and hsa-miR-378a-5p are significantly downregulated in tumors of the TCGA-COAD and TCGA-READ cohorts (Figure 3F,G); however, age-specific TCGA analysis revealed upregulation in young (<50 years) tumor datasets (Figure 4F,G). EOCRC-specific expression of miRNAs seems to differ from the overall miRNA profile, with a larger number of LOCRC cases skewing the data to resemble that of late-onset CRC.

Upregulation/downregulation observed in the age-specific TCGA analysis confirmed our previous RNA-Seq analysis for all miRNAs, except hsa-miR-326, hsa-miR-378a-5p, hsa-miR-378c and hsa-miR-378d. These miRNAs were found to be upregulated in young tumors of the TCGA cohort, whereas our RNA-Seq analysis showed downregulation in young patients. Hence, to resolve the anomalies between TCGA data and our RNA-seq results, we needed to validate our observations in additional cohorts. For downstream validation by RT-qPCR, we left out hsa-miR-378e as TCGA analysis revealed no expression in young datasets. Selected miRNAs were further validated in additional EOCRC and LOCRC cohorts of 16 young patients (<50 years old) and 11 old patients (>50 years old). Significantly upregulated DEMs included hsa-miR-1247-3p and hsa-miR-148a-3p. Hsa-miR-326 was significantly downregulated in both EOCRC discovery and validation cohorts in contrast to upregulation in TCGA-COAD and TCGA-READ young tumor samples. Additionally, the slight change in hsa-miR-326 expression between young and old normal tissues observed in the TCGA age-specific analysis was not replicated in the RT-qPCR validation cohort. The possible reason behind this could be that the ethnicity of the population in TCGA cohorts consists of Caucasians and African Americans, with no representation of the Indian population amongst them [44]. Hsa-miR-326 seems to potentially show ethnicity-specific changes accounting for the difference in expression between TCGA datasets and our Indian validation cohort. Since our study concentrates on an East and North Indian population, racial differences may contribute to this variation in hsa-miR-326 expression.

Experimentally validated targets of the selected miRNAs were compared with differentially expressed (DE) genes of the TCGA-COAD and TCGA-READ cohorts to identify predicted DE-miRNA (DEM) targets altered in colorectal tumor tissue in a direction reciprocal to that of the miRNAs. Most of the DE target genes of upregulated miRNA hsa-miR-1247-3p (a total of 20 in number) were enriched in biological processes of anatomical structure morphogenesis (10/20 genes), receptor-mediated endocytosis (4/20 genes), and in molecular function of phosphatidylinositol-3,5-bisphosphate binding (2/20 genes). DE target genes of upregulated miRNA hsa-miR-148a-3p (total 26 in number) were enriched in biological processes of negative regulation of metabolic processes (11/26 genes), regulation of phosphorus metabolic processes (8/26 genes), tissue morphogenesis (7/26 genes), cellular response to nutrient levels (5/26 genes) and negative regulation of anoikis (3/26 genes). Cellular component analysis of miR-148a-3p DE-target genes revealed enrichment of genes located in the ruffle membrane (3/26 genes), which is an indicator of tumor cell motility and metastatic ability [45]. The overall picture indicates a dysregulation of metabolic pathways and deregulated tissue morphogenesis contributing to epithelial-mesenchymal plasticity. Tumor cells are able to meet the demands of enhanced growth and proliferation by a plethora of metabolic reprogramming and also by competing with other surrounding cells and consuming essential nutrients from the microenvironment [46,47,48]. Deregulated tissue morphogenesis has important physiological significance, as the downregulation of epithelial gene expression signature and the dissolution of epithelial intercellular junctions are key events in epithelial-mesenchymal transition (EMT) [49]. Negative regulation of anoikis molecular pathways promotes anchorage-independent growth and EMT, leading to cancer progression and tumor metastasis [50].

Downregulated miRNA hsa-miR-326 DE-target genes (a total of 32 in number) were found to be enriched mostly in the biological processes of vasculature development/blood vessel development (7/32 genes) and gland development (6/32 genes). These predicted upregulated targets include various molecules whose expression are known to correlate with the parameters of disease aggressiveness like tumor invasion, angiogenesis, liver metastasis, disease recurrence, and poor prognosis [51,52,53,54,55,56,57,58,59,60,61,62,63,64]. Cellular component analysis of upregulated DE-targets indicated enrichment in the basolateral plasma membrane (3/32 genes) or the basal part of the cell. Intestinal epithelial cells are known to exhibit epithelial cell polarity with distinct apical and basolateral plasma membrane domains [65]. The basolateral plasma membrane is rich in phosphatidylinositol-3,4,5-trisphosphate and contains junctional complexes that regulate intercellular adherence and adherence with the basement membrane [65]. Alterations of basolateral membrane proteins have been found to correlate with loss of epithelial architecture and onset of cancer [66]. Taken together, our results potentially indicate metabolic reprogramming, deregulation of anoikis-regulating pathways, and alterations in proteins present in the basal part of intestinal epithelial cells.

In spite of the bulk of previously conducted studies on CRC gene expression and miRNAs [13,14,27,29,67,68,69,70], there are significant gaps in our understanding of EOCRC and how it differs from LOCRC. Most of the published reports with genome-wide RNA sequencing [14,67] concentrate on a cohort comprised exclusively of EOCRC patients. The inclusion of sporadic LOCRC is essential in the initial discovery cohort, as it is not possible to identify markers specifically deregulated in early-onset disease (significantly upregulated/downregulated with respect to LOCRC) without comparison with LOCRC tissues. The diagnosis of CRC before the age of 50 always raises the suspicion of a genetic cancer predisposition syndrome (Lynch syndrome and familial adenomatous polyposis). So, it is important to screen the EOCRC/LOCRC cohort for known canonical markers as their presence creates a hypermutable and pro-oncogenic phenotype. Liu et al. performed genome-wide miRNA and transcriptome profiling, but the tumors included in their study cohorts were not assessed for canonical CRC markers like MSI activation of the Wnt pathway, or *APC* mutations [29]. Another bottleneck of big-data transcriptomic studies is that they very rarely include paired adjacent colonic mucosa as normal samples in their analysis. Ideally, tumor samples should be paired with corresponding normal samples to avoid biological differences between individuals.

Our study is the first attempt to identify differentially expressed miRNAs specific to EOCRC in the Indian population. We have attempted to remove the inconsistencies of previous studies by the inclusion of a LOCRC cohort, both in the discovery cohort and also in the validation cohort, with paired adjacent colonic mucosa corresponding to each tumor analyzed in this study. One limitation of our study is the small sample size of the discovery cohort. To compensate for that, we have validated our findings in the TCGA-COAD and TCGA-READ datasets and finally in an additional cohort of 16 young patients and 11 aged patients. Other than hsa-miR-326, all selected miRNAs showed similar expression between age-specific TCGA datasets and our study cohorts. Additionally, we have also screened our cohorts for MSI, nuclear β-catenin, and *APC* mutations to collect tumors negative for these known CRC canonical markers. To the best of our knowledge, our study is the first study that incorporates TCGA-COAD and TCGA-READ-based age-specific validation along with RT-qPCR to identify miRNAs deregulated in early-onset CRC. Since a miRNA can target many mRNAs, we screened the targets of our EOCRC-validated DEMs to identify those target genes (46 downregulated and 32 upregulated) known to be differentially expressed in colorectal adenocarcinoma (TCGA-COAD and TCGA-READ) datasets. In the future, these target genes need to be explored in EOCRC and LOCRC cohorts for the identification of potential pathways responsible for the early onset of colorectal cancer.

## 4. Materials and Methods

### 4.1. Patient Recruitment

34 (7 patients < 50 years, 27 patients > 50 years) colorectal tumor samples and respective adjacent normal colonic mucosas from histologically proven CRC patients were collected in collaboration with doctors and pathologists from the Surgical Oncology Department of Netaji Subhas Chandra Bose Cancer Hospital, Kolkata (discovery cohort). This cohort was used for screening MSI, nuclear β-catenin, and *APC* mutations. 5 (3 patients < 50 years, 2 patients > 50 years) colorectal tumors and adjacent normal colonic mucosas (total 10 tissue samples) sent for small-RNA seq were screened from this cohort. 56 (24 patients < 50 years, 32 patients > 50 years) histologically proven colorectal tumors and respective adjacent normal colonic mucosas were obtained from the Departments of Surgical, Medical and Radiation Oncology, Surgical Gastroenterology and General Surgery, All India Institute of Medical Sciences (AIIMS), Rishikesh (validation cohort). 27 (16 patients < 50 years, 11 patients > 50 years) tumors and adjacent normal colonic mucosas were screened from this cohort for validating our RNA-seq results. All patients with histopathologically proven colorectal tumors undergoing treatment from May 2020 to May 2022 who fulfilled the inclusion and exclusion criteria were included. All necessary IEC permissions were obtained prior to sample collection. Clinicopathological information like age, sex, site, stage, and differentiation of tumor, familial history of CRC, and presence of any other inflammatory bowel disease was also collected. Inclusion criteria: Patients admitted for surgical resection with biopsy-proven colorectal adenocarcinoma, age up to 80 years, willing to provide written informed consent. Exclusion criteria: Patients with Familial Colorectal Carcinoma, Unable/unwilling to give consent, patients with cancers other than CRC, and patients receiving neoadjuvant chemotherapy and/or radiotherapy.

### 4.2. Biospecimen Collection

Tumor and normal tissue samples were collected in RNALater (RNAlater^TM^ Stabilization Solution, Invitrogen, catalog# AM7020, Carlsbad, CA, USA) and 10% neutral buffered formalin. Samples collected in RNALater (Invitrogen) for nucleic acid extraction were stored at −80 °C for processing at a later date. Samples stored in neutral buffered formalin were processed into FFPE blocks, sectioned into 5 µm sections, and adhered onto positively charged slides. Histological sections were stained with hematoxylin and eosin. All specimens with histopathological features suggestive of an inflammatory colorectal disease were excluded from this study. Reporting was performed by trained histopathologists. Grossing and reporting of colectomy specimens suspicious of colorectal carcinoma were conducted according to CAP (College of American Pathologists).

### 4.3. Immunohistochemistry

IHC for MMR proteins (MLH1, MSH2, MSH6, and PMS2) and nuclear β-catenin was performed as per standard protocol. Briefly, about 5–10 μm paraffin sections of tissue samples were deparaffinized and rehydrated in a series of graded alcohols. Heat-induced antigen retrieval was conducted in Tris-EDTA buffer (pH 9) for MMR proteins and in 10 mM sodium citrate buffer (pH 6) for nuclear β-catenin in the microwave, followed by peroxidase quenching and antibody blocking with 3% BSA. Slides were then subjected to overnight incubation at 4 °C with the respective primary antibodies at standardized dilutions (given in Table 5). The slides were developed using 3-3′ diaminobenzidine (DAB) as the chromogen and counterstained with hematoxylin. The PolyExcel HRP/DAB detection system—TWO STEP Universal kit for Mouse and Rabbit primary antibodies (PathnSitu, catalog# PEH002, Pleasanton, CA, USA), was used for the qualitative identification of the nuclear antigens. Slides were analyzed by a trained histopathologist for the detection of MSI (as per CAP guidelines) and nuclear β-catenin. Normal colorectal tissue was taken as an internal control. A known case of MSI CRC was used as a positive control for MSI detection. External control for nuclear β-catenin consisted of a histologically diagnosed section of fibromatosis (desmoid tumor). No antibody controls were taken as negative controls. The scoring of Wnt positive nuclear β-catenin expression (Wnt+) was performed according to Raman et al. [12]. A sample was scored as Wnt positive (Wnt+) if the β-catenin nuclear stain was observed in more than 35% of tumor epithelial cells and Wnt negative (Wnt−) if a nuclear stain was detected in less than 25% of cells. IHC images were captured using the Olympus BX53F2 (Olympus, model# BX53F2, Tokyo, Japan) biological microscope.

### 4.4. RNA Isolation

RNA isolation was performed from fresh frozen colorectal (tumor and normal) tissues stored in RNALater (Invitrogen) at −80 °C. Diethyl Pyrocarbonate (DEPC) treatment of glassware and forceps was performed prior to RNA isolation. Isolation was performed using the Qiagen AllPrep^®^ DNA/RNA/miRNA Universal kit (Qiagen, catalog# 80224, Hilden, Germany) as per the manufacturer’s protocol. Elution was performed in nuclease-free water. For isolation of RNA from tissues in the validation cohorts, 50–100 mg of paired tumor and adjacent normal colonic tissues were chopped into small pieces using a sterile surgical scalpel. The tissues were then homogenized in 1 mL of TRIzol reagent (Invitrogen, catalog# 15596026) and incubated at 4 °C overnight for efficient homogenization and lysis. Downstream processing for RNA isolation from TRIzol reagent (Invitrogen) was conducted as per standard protocol.

### 4.5. DNA Isolation from Tissue

DNA isolation from tissue was performed by standard protocol. Chopped 50 mg tissue samples were incubated in digestion buffer (60 mM Tris pH 8.0, 100 mM EDTA, 0.5% SDS) and proteinase K (500 ng/mL) at 56 °C overnight. An equal volume of phenol-chloroform solution was added and mixed well by inverting repeatedly. Tubes were centrifuged at 12,000× *g* for 15 min at room temperature for phase separation. An equal amount of chloroform was added and mixed well before centrifugation at 12,000× *g* for 15 min at room temperature for phase separation. 1/10th volume of 3M sodium acetate pH 6.0 and 2.5 volumes of absolute ethanol were added to the aqueous phase and kept at −20 °C overnight for nucleic acid precipitation. Centrifugation was performed at 12,500× *g* for 15 min at 4 °C and washed with 70% ethanol at 12,500× *g* for 5 min at 4 °C. The DNA pellet was air-dried and resuspended in 100 µL of TE pH 8.0. RNase treatment (20 µg/mL) was conducted for 30 min at room temperature to eliminate RNA. Phenol-chloroform phase separation and chloroform phase separation steps were repeated to remove the RNase and then DNA was purified from the aqueous phase by using Promega kit protocol (Promega Wizard SV Gel and PCR Clean-Up System, part# 9FB072, Madison, WI, USA). Isolated DNA was quantitated by spectrometry and visualized by electrophoresis in 0.8% agarose gel using the BioRad GelDoc imaging system (Bio-Rad Laboratories, Hercules, CA, USA).

### 4.6. PCR for APC Gene

Normal and tumor DNA from each patient was subjected to PCR amplification. Primer sequences were designed to amplify the mutation cluster region (exon 15, codons 1260–1596) of the APC gene in overlapping PCR segments. Reaction conditions were as follows: 95 °C, 5 min; 95 °C, 30 s; 60 °C/57 °C/60 °C (Annealing Temperature—T_a_), 1 min; 68 °C, 1 min; 68 °C, 5 min; for 35 cycles.

PCR amplification was performed in a 25 µL reaction with 1 unit of NEB Taq DNA polymerase (NEB, catalog# M0273S, Ipswich, MA, USA), 1X Standard Taq Buffer (NEB), 200 µM dNTPs (NEB), 1µM forward and reverse primer each (Table 6) and 100 ng of template DNA. The PCR product was visualized by electrophoresis in a 2% agarose gel in 1X TBE (0.13M Tris (pH 7.6), 45mM boric acid, 2.5mM EDTA) buffer.

Information on PCR primers is provided below (Table 6):

### 4.7. Direct DNA Sequencing

The PCR products were purified using the Promega Wizard SV Gel and PCR Clean-Up System (Promega) according to the manufacturer’s instructions. Direct sequencing was performed by Eurofins Genomics India Pvt. Ltd. (Whitefield, Bangalore, India).

### 4.8. APC Gene Mutation Analysis

The nucleotide and deduced amino acid sequences were compared with reference sequences of the APC gene available at the NCBI (National Center for Biotechnology Information) GenBank database using the BLASTx (Basic Local Alignment Search Tool) program.

### 4.9. miRNA Seq Analysis

A total of 2 μg RNA samples isolated from 10 tissue types (5-tumor, 5-normal) were sent for RNA sequencing. RNA samples were outsourced to the company Bencos Research Solutions (Kolkata, India) for RNA sequencing and data analysis. RNA with RIN > 8.0 proceeded for library preparation using the NEBNext^®^ Small-RNA Library Prep Set for Illumina^®^ (Illumina Inc., San Diego, CA, USA) and sequenced at the National Genomics Core, CDFD, Hyderabad, using 50-bp single-end reads on Illumina NextSeq 500 Sequencer (Illumina Inc.). The data was generated by using the paired-end approach of the Illumina technique. A total of twenty paired-end fastq files were used for the small-RNA-seq analysis via a pipeline of FastQC-FastpSortMeRNA-miRDeep2-edgeR. Fastq files were subjected to Fastqc (v.0.11.9) for the sequence quality check (FastQC; https://www.bioinformatics.babraham.ac.uk/projects/fastqc/, accessed on 14 January 2022) and found that all the quality features were passed, except some features that were flagged with warnings and failed. After sequencing, adapters, and low-quality sequences were removed from the obtained raw reads by the Fastp tool (v.0.23.2) [71], and clean data was counted using the FastQC program. All the reads of every sample were subjected to SortMeRNA (v.4.3.2) for removal of ribosomal RNA sequences [72]. Four databases, silva-euk-28s-id98, silva-euk-18s-id95, rfam-5.8s-database-id98, and rfam-5s-database-id98 were utilized in the rRNAs removal analysis (https://github.com/biocore/sortmerna/archive/2.1.tar.gz, accessed on 20 January 2022). Ten independent read mappings of small-RNA-seq data were performed by miRDeep2 (v.2.0.1.2) using a genome reference sequence database Homo_sapiens. GRCh38.dna.primary_assembly.fa [73] along with mature and hairpin miRNA sequences specific to humans (hsa as a species) retrieved from the miRBase (v22.1) database (https://www.mirbase.org, accessed on 22 January 2022). The reference sequence fasta file was downloaded from the Ensembl genome base (http://ftp.ensembl.org/pub/release-105/fasta/homo_sapiens/dna/, accessed on 22 January 2022).

### 4.10. Identification of Known and Novel miRNAs

Processed reads were used to generate collapsed reads using mapper.pl module of the miRDeep2 package with a minimum length of 18 parameters. For predicting miRNAs, known and novel, the collapsed reads were passed to miRDeep2.pl module of the package. In this analysis, the reference genome sequences and the miRBase mature and hairpin sequences specific to humans (hsa as a species) were utilized. The count matrix was generated using the final results after removing the duplicates based on the same genomic coordinates.

### 4.11. Differential miRNA Expression Analysis

Read count was performed for all the samples using miRDeep2. For differential miRNA expression analysis, edgeR (v.3.36.0) R-package was utilized [74]. The normalization factor was calculated using raw read counts, and after that, the count data were normalized by the Count Per Million (CPM) method. The exact test method is available in the egdeR package, which was implemented for the differential expression analysis (DEA) of the single sample comparisons. In the single sample comparisons, a 0.2 divergence value was considered for the DEA. However, the lmFit module of edgeR, a linear model using weighted least squares for each gene, was utilized to fit the linear model into the data for multiple sample comparisons after applying the voom transformation and calculation of variance weights. Further, the empirical Bayes (eBayes) module of edgeR was employed for smoothing the standard errors and calling the differentially expressed transcripts of miRNAs. Differentially expressed miRNAs were obtained by filtering the results of DEA using adjusted *p*-value or FDR ≤ 0.05 and Log_2_ fold change value at ≥2 (upregulation) or ≤−2 (downregulation).

### 4.12. Volcano Plots

Global gene expression values obtained from a pairwise comparison analysis were also plotted in the form of a volcano plot using the R-package. In the volcano plots, miRNA rows were ordered from the final result of edgeR analysis according to the adjusted *p*-value or FDR in decreasing order. The volcano plot was generated for all five single-sample comparisons.

### 4.13. Heatmap

The pheatmap package in R was implemented to plot the heatmaps, and the rlog-normalized read count matrix for all the samples was used as the input data (https://cran.r-project.org/web/packages/pheatmap/index.html, accessed on 23 May 2023) derived from the DESeq2 R-package [75]. The function rlog returns a Summarized Experiment object that contains the rlog-transformed values in its assay slot. Corresponding Z-scores were computed from the rlog-normalized read count matrix, and pheatmap drew the heatmap accordingly. The top 20 most variable miRNAs were extracted from the matrix to be plotted on the heatmap. The miRNAs that showed expression values higher than the mean expression across samples were assigned a positive Z-score denoted by green. The opposite, that is, the negative Z-score is denoted by a red on the heatmap.

### 4.14. Data Acquisition and Processing from TCGA-COAD and TCGA-READ Database

In order to obtain the expression values of miRNAs, expressed in different ages of normal and tumor samples of COAD and READ patients, we downloaded and processed the bulk RNA seq data along with clinical files of TCGA-COAD and TCGA-READ project by using “*TCGAbiolinks*“ R package (version 2.19.2) [76]. With the “*TCGAbiolinks*“, the RNAseq raw count matrix was downloaded from the GDC server. By using a few more R packages tidyr, dplyr, tibble [77,78,79] we extracted the miRNAs of our interest and their expression values for four groups:

(a) miRNAs found in normal samples having age > 50 years we grouped them as N_O samples (nCOAD = 165; nREAD = 45), (b) miRNAs found in normal samples having age < 50 years we grouped them as N_Y samples (nCOAD = 32; nREAD = 15), (c) miRNAs found in tumor samples having age > 50 years we grouped them as T_O samples (nCOAD = 183; nREAD = 83), (d) miRNAs found in tumor samples having age < 50 years we grouped them as T_Y samples (nCOAD = 22; nREAD = 5). ‘nCOAD’ represents the sample size for COAD dataset, ‘nREAD’ represents the sample size for READ dataset. miRNA expression values of all four groups (Normal_Old, Normal_Young, Tumor_Old, Tumor_Young) for COAD and READ datasets are given in Appendix A.

### 4.15. Target Identification and Selection

Experimentally validated targets for the selected miRNAs were identified using miRNet [37]. Additionally, genes differentially expressed (DE) in TCGA colon adenocarcinoma (COAD) and TCGA rectal adenocarcinoma (READ) samples with |Log_2_FC| cutoff of 1.00 and q-value cutoff of 0.01 were derived using GEPIA [38]. Target genes were compared with this DE-gene list to identify DE-targets altered in colorectal tumor tissue in a direction reciprocal to that of miRNAs. Identities of DE genes and DE targets for each miRNA are given in Appendix A.

### 4.16. Gene Ontology and Pathway Enrichment Analysis

For each comparison group, gene ontology (GO) and pathway enrichment analysis were performed separately for both the upregulated and downregulated sets of differentially expressed miRNAs’ target genes against humans as a selected species. A tool, ShinyGO (v.0.81) [40], was used for retrieving functional annotations based on the Biological Process (BP), Molecular Function (MF), and Cellular Component (CC). FDR was calculated based on the nominal *p*-value from the hypergeometric test. Identities of miRNA targets differentially expressed in TCGA-COAD and TCGA-READ and enriched in GO Enrichment pathways are given in Appendix A.

### 4.17. miRNA Validation by RT-qPCR

Real-time analyses by two-step RT-qPCR were performed for quantification of miRNA levels. The stem-loop RT-qPCR method was used for miRNA screening and quantification [35,36]. Reverse transcription (RT) was performed with Verso cDNA Synthesis Kit (Thermo Scientific, catalog #AB-1453/A, Van Allen Way, Carlsbad, CA, USA) as per the manufacturer’s instructions using 100 ng of total cellular RNA. The 10 µL of RT reaction mixture contained 1 µL of RT primer (1 µM), 500 µM each of dNTP, 2 µL of 5X cDNA synthesis buffer, 0.5 µL of RT enhancer, and 0.5 µL of Verso Enzyme Mix (Thermo Scientific). All miRNA RT-qPCRs were performed on the Biorad CFX96 Real-Time System (Bio-Rad Laboratories, Hercules, CA, USA). One-tenth of the reverse transcription mix was subjected to PCR amplification with Bio-Rad SsoAdvanced Universal SYBR^®^ Green Supermix (Bio-Rad, catalog #1725270). The 20 µL of RT-qPCR reaction mixture contained 2 µL of forward and reverse primers (1 µM each) and 10 µL of 2X SsoAdvanced Universal SYBR^®^ Green Supermix (Bio-Rad). The RT reaction condition was: 25 °C, 10 min; 42 °C, 60 min; 95 °C, 5 min; 4 °C, ∝. The RT-qPCR condition was: 95 °C, 3 min; 95 °C, 30 s; 60 °C, 1 min; for 40 cycles. All Samples were analyzed in triplicates. The concentrations of intracellular miRNAs were calculated based on their normalized Ct values. Normalization was performed by U6 snRNA. The ΔΔCt method for relative quantitation (RQ) of gene expression was used and relative quantification was performed using the equation 2^−ΔΔCt^ (as per ‘Guide to Performing Relative Quantitation of Gene Expression Using Real-Time Quantitative PCR’ by Applied Biosystems (https://assets.thermofisher.com/TFS-Assets/LSG/manuals/cms_042380.pdf, accessed on 30 December 2024) [80]. Briefly, ∆Ct = Ct_miRNA_–Ct_U6_, ∆∆Ct = ∆Ct_TUMOR/NORMAL_–∆Ct_NORMAL_, 2^−∆∆Ct^ represents the relative quantification as compared to the respective normal.

Information on miRNA Reverse Transcription (RT) Stem Loop Primers (SLP), miRNA Real-time PCR forward and reverse primers are provided below in Table 7 and Table 8 respectively.

The reverse primer sequence is complementary to a portion of the RT USLP and is hence common for all miRNAs.

### 4.18. Statistical Analysis

All graphs were plotted and analyzed using GraphPad Prism 8.00 (GraphPad, San Diego, CA, USA). For statistical analysis, a non-parametric two-tailed, paired, or unpaired Student’s t-test was performed. Error bars indicate mean with standard deviation.

## Figures and Tables

**Figure 1 ncrna-11-00010-f001:**
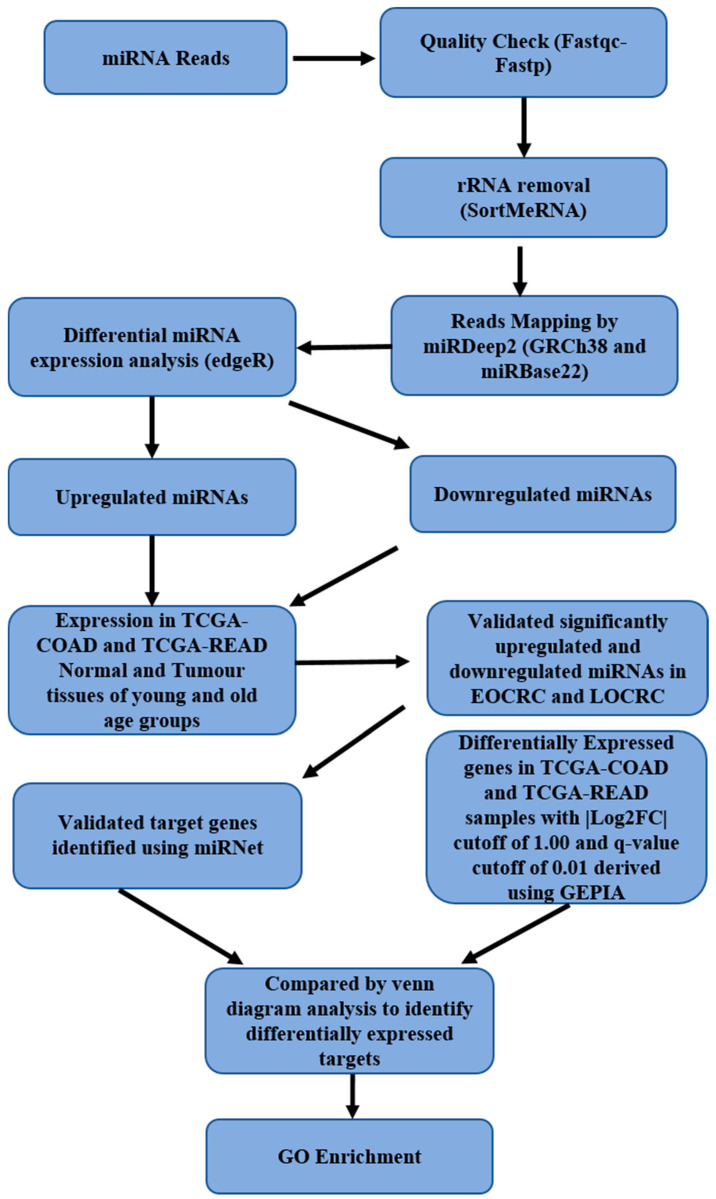
**Schematic workflow of overall pipeline used for the small RNA (miRNA) transcriptome analysis.** Schematic work-flow of overall pipeline used for the small-RNA (miRNA) transcriptome analysis of sequencing data generated from 5 (3 = EOCRC, 2 = LOCRC) normal colonic mucosas and 5 (3 = EOCRC, 2 = LOCRC) colorectal tumors. A total of twenty paired-end fastq files were used for the small-RNA-seq analysis via a pipeline FastQC-FastpSortMeRNA-miRDeep2-edgeR. Top differentially expressed miRNAs were validated first in TCGA-COAD and TCGA-READ age-specific groups and then in additional EOCRC and LOCRC cohorts. Validated targets of significantly differentially expressed miRNAs were compared with differentially expressed genes in TCGA-COAD and TCGA-READ datasets to identify potential targets deregulated in EOCRC.

**Figure 2 ncrna-11-00010-f002:**
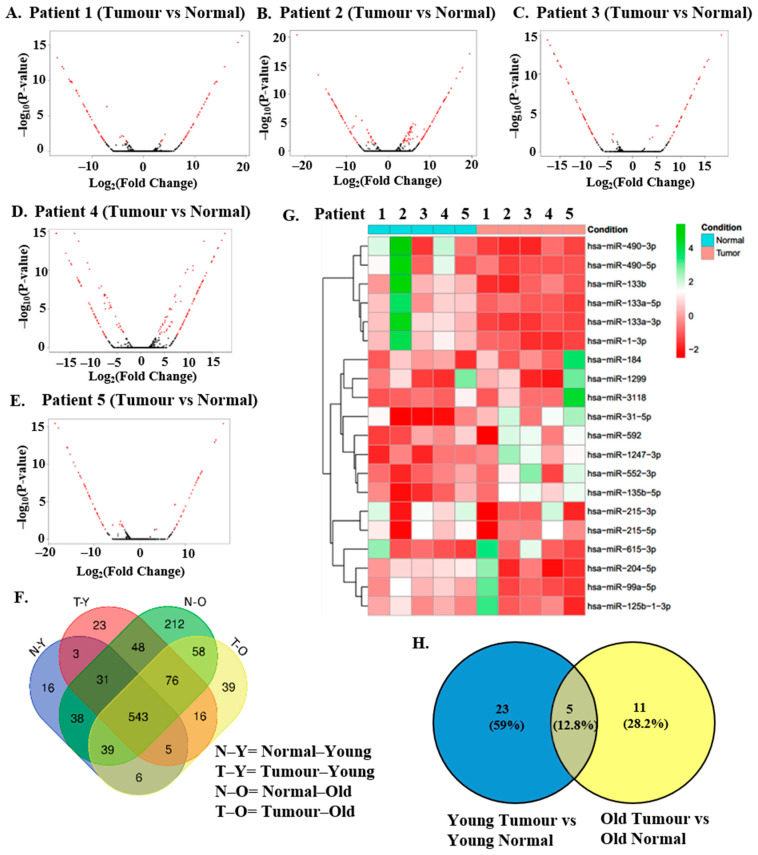
**Identification of Differentially Expressed miRNAs (DEMs) in tumor samples of the discovery cohort.** (**A**–**E**): Significant differentially expressed miRNAs were identified by filtering the results with a cut-off parameter adjusted *p*-value or FDR < 0.05. Global miRNA expression values obtained from a pair-wise comparison (tumor miRNA compared with corresponding normal) analysis for each patient in the discovery cohort were plotted in the form of volcano plots. Differentially expressed miRNAs at the level of statistical significance adjusted *p*-value (FDR) < 0.05 are shown in red-colored dots, while non-significant miRNAs are in black-colored dots. Y-axis shows the negative logarithm of FDR values, and x-axis represents log fold change value of each transcript. Patients 1, 2, and 3 represent young patients <50 years old, and Patients 4 and 5 represent old patients >50 years old. (**F**) All identified miRNAs of Normal_old, Tumor_old, Normal_young, and Tumor_young were used for the generation of Venn diagram. The Venn diagram was constructed via Venn (https://bioinformatics.psb.ugent.be/webtools/Venn/, accessed on 17 March 2022). (**G**) Heatmap of top 20 miRNAs sorted by variance in decreasing order, found across 10 samples. The heatmap was generated by using the rlog-normalized count data of each sample. Z-scores for each miRNA per sample are indicated in the red and green color scale shown on the right for expression levels below and above mean expressions across the sample for that given miRNA, respectively. The condition legend denotes the sample types, where cyan blue is assigned to normal samples and pink is assigned to tumor samples. (**H**) Venn diagram shows known DEMs with cut-off *p*-value < 0.05 and Log_2_ fold change value at >2 or <−2 identified in the multiple sample comparisons of Young Tumor vs. Normal and Old Tumor vs. Normal (normal as control group and tumor as test group)**.** Constructed using https://bioinfogp.cnb.csic.es/tools/venny/, accessed on 2 June 2022.

**Figure 3 ncrna-11-00010-f003:**
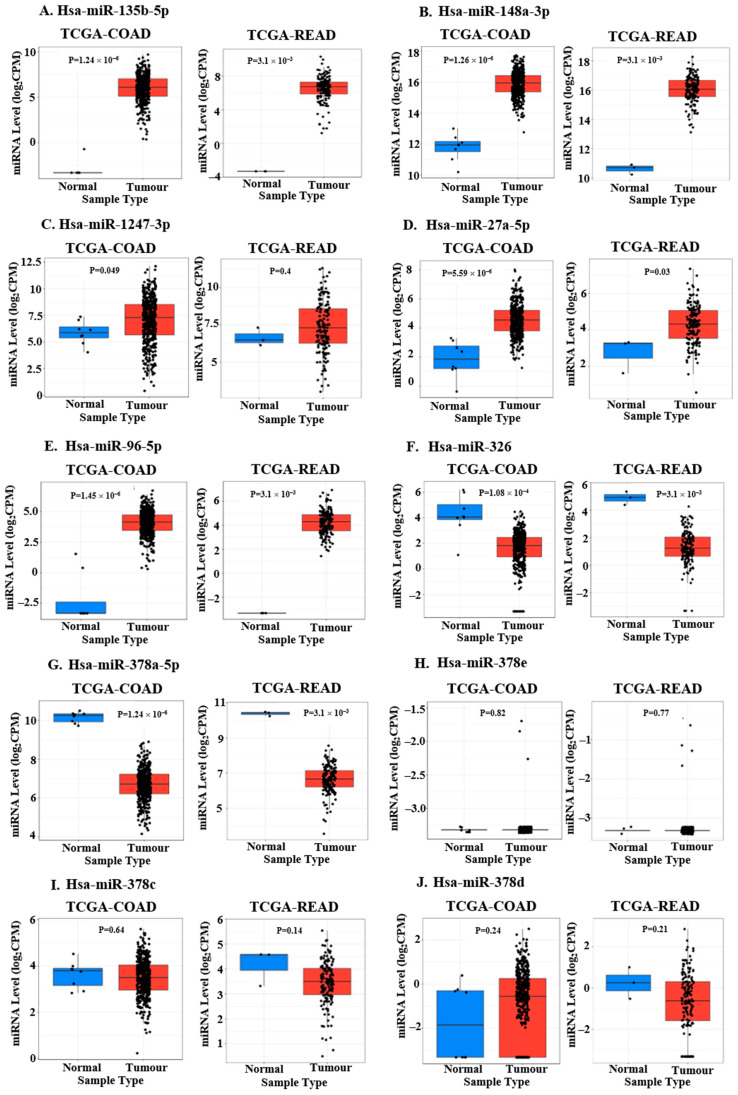
TCGA-based validation of top 10 miRNAs differentially expressed in sample comparison of young tumors with young normal tissues on Cancer MIRNome (CancerMIRNome (jialab-ucr.org) COAD—accessed on 12 June 2022, READ—accessed on 25 December 2024). Gene expression analysis of (**A**) hsa-miR-135b-5p, (**B**) hsa-miR-148a-3p, (**C**) hsa-miR-1247-3p, (**D**) hsa-miR-27a-5p, (**E**) hsa-miR-96-5p, (**F**) hsa-miR-326, (**G**) hsa-miR-378a-5p, (**H**) hsa-miR-378e, (**I**) hsa-miR-378c, and (**J**) hsa-miR-378d on the TCGA datasets of (left panel) colon (COAD) and (right panel) rectum adenocarcinomas (READ).

**Figure 4 ncrna-11-00010-f004:**
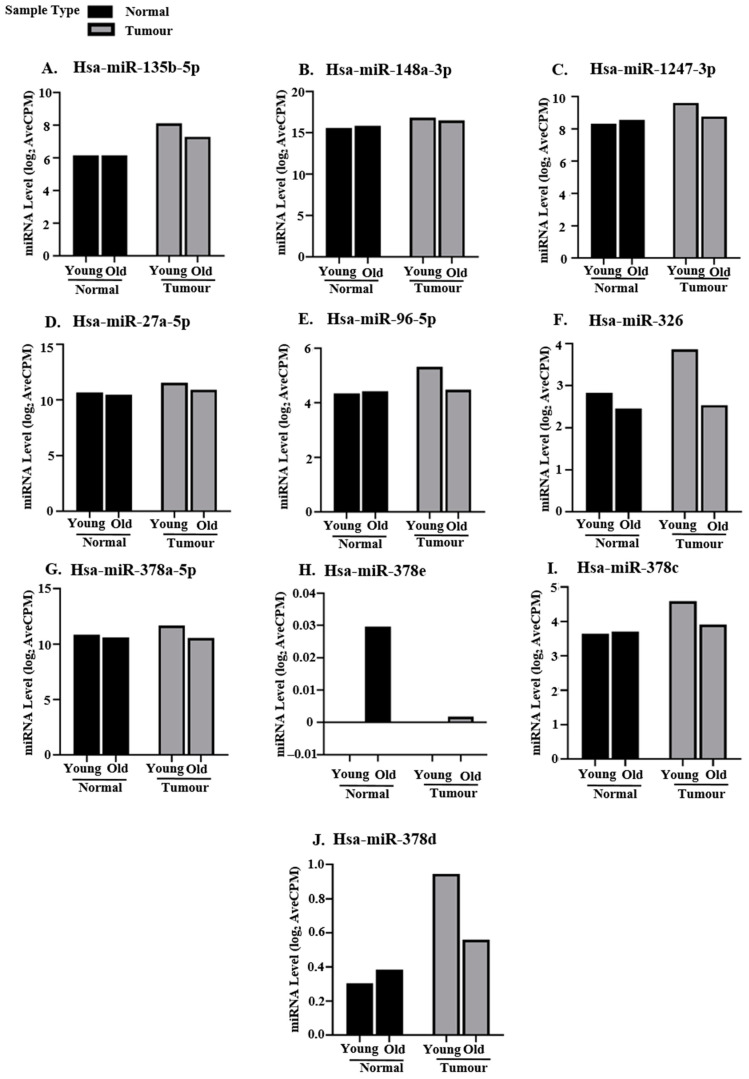
Comparative expression of top 10 miRNAs in Young Normal, Old Normal, Young Tumor, and Old Tumor populations of the TCGA-COAD and TCGA-READ datasets. Log_2_AveCPM of miRNA counts was calculated by averaging the CPM of selected miRNAs in COAD and READ datasets for Young Normal, Old Normal, Young Tumor, and Old Tumor. miRNA levels (log_2_AveCPM) of (**A**) hsa-miR-135b-5p, (**B**) hsa-miR-148a-3p, (**C**) hsa-miR-1247-3p, (**D**) hsa-miR-27a-5p, (**E**) hsa-miR-96-5p, (**F**) hsa-miR-326, (**G**) hsa-miR-378a-5p, (**H**) hsa-miR-378e, (**I**) hsa-miR-378c, and (**J**) hsa-miR-378d in Young Normal (*n* = 47; nCOAD = 32, nREAD = 15), Old Normal (*n* = 210; nCOAD = 165, nREAD = 45), Young Tumor (*n* = 27; nCOAD = 22, nREAD = 5), and Old Tumor (*n* = 266; nCOAD = 183, nREAD = 83) populations of the TCGA datasets of colon and rectum adenocarcinomas (COAD and READ). ‘n’ represents the total sample size. ‘nCOAD’ represents the sample size for TCGA-COAD dataset, and ‘nREAD’ represents the sample size for TCGA-READ dataset.

**Figure 5 ncrna-11-00010-f005:**
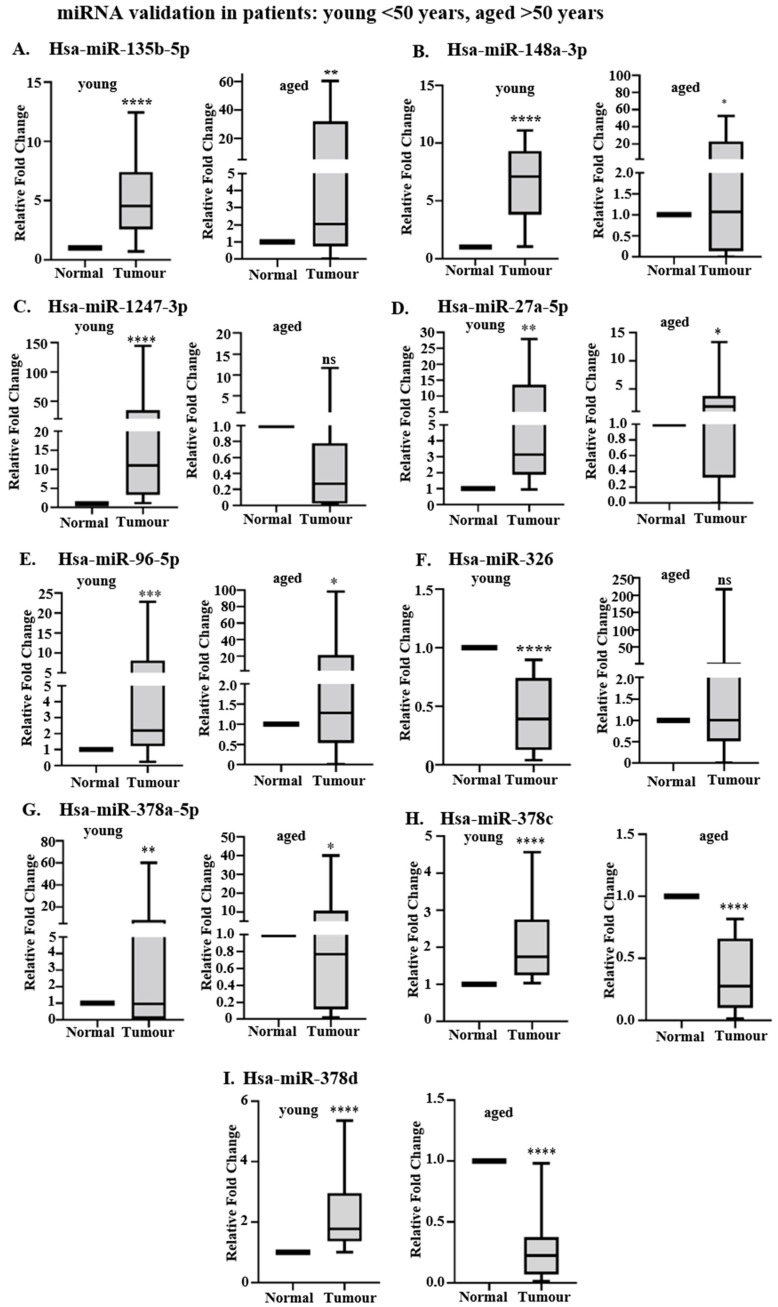
RT-qPCR-based differential analysis of TCGA-validated miRNAs in additional EOCRC and LOCRC validation cohorts. Relative expression analysis of (**A**) hsa-miR-135b-5p, (**B**) hsa-miR-148a-3p, (**C**) hsa-miR-1247-3p, (**D**) hsa-miR-27a-5p, (**E**) hsa-miR-96-5p, (**F**) hsa-miR-326, (**G**) hsa-miR-378a-5p (**H**) hsa-miR-378c, and (**I**) hsa-miR-378d in tumors and adjacent normal tissues of 16 young patients (<50 years old) of our EOCRC (left panel) and 11 aged patients (>50 years old) of our LOCRC (right panel) validation cohorts. RT-qPCR detection of miRNAs was conducted from 200 ng of isolated cellular RNA. N = 3 technical replicates for each patient tissue (tumor and normal). Normalization was performed by U6 snRNA. Data represents Mean ± SD. Statistical significance was calculated by two-tailed Students’ paired *t*-test. ns: non-significant, ‘*’ denotes statistically significant *p*-value, * means *p* < 0.05, ** means *p* < 0.01, *** means *p* < 0.001. **** means *p* < 0.0001.

**Figure 6 ncrna-11-00010-f006:**
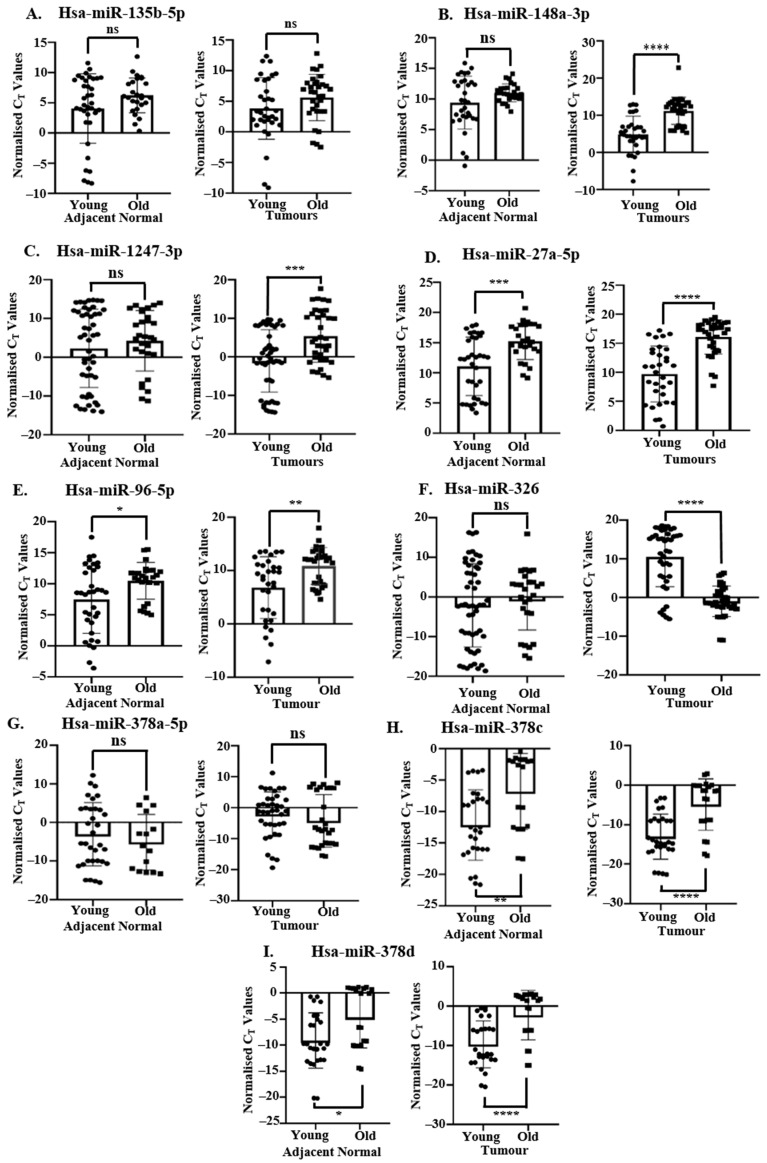
RT-qPCR-based differential analysis of validated DEMs in young and old tissues in additional EOCRC and LOCRC validation cohorts. Normalized Ct values (Ct_miRNA_–Ct_U6_) of miRNAs in adjacent normal tissues (left panel) and tumors (right panel) were compared between young (EOCRC validation cohort) and old patients (LOCRC validation cohort). Relative normalized Ct values of (**A**) hsa-miR-135b-5p, (**B**) hsa-miR-148a-3p, (**C**) hsa-miR-1247-3p, (**D**) hsa-miR-27a-5p, (**E**) hsa-miR-96-5p, (**F**) hsa-miR-326, (**G**) hsa-miR-378a-5p, (**H**) hsa-miR-378c, and (**I**) hsa-miR-378d in adjacent normal (left panel) and tumor tissues (right panel) of 16 young patients and 11 old patients, N = 3 technical replicates for each. Normalized Ct = Ct_miRNA_–Ct_U6_. RT-qPCR detection of miRNAs was performed from 200 ng of isolated cellular RNA. Normalization was conducted by U6 snRNA. Data represents Mean ± SD. Statistical significance was calculated by two-tailed Students’ unpaired t-test. ns: non-significant, ‘*’ denotes statistically significant *p*-value, * means *p* < 0.05, ** means *p* < 0.01, *** means *p* < 0.001. **** means *p* < 0.0001.

**Figure 7 ncrna-11-00010-f007:**
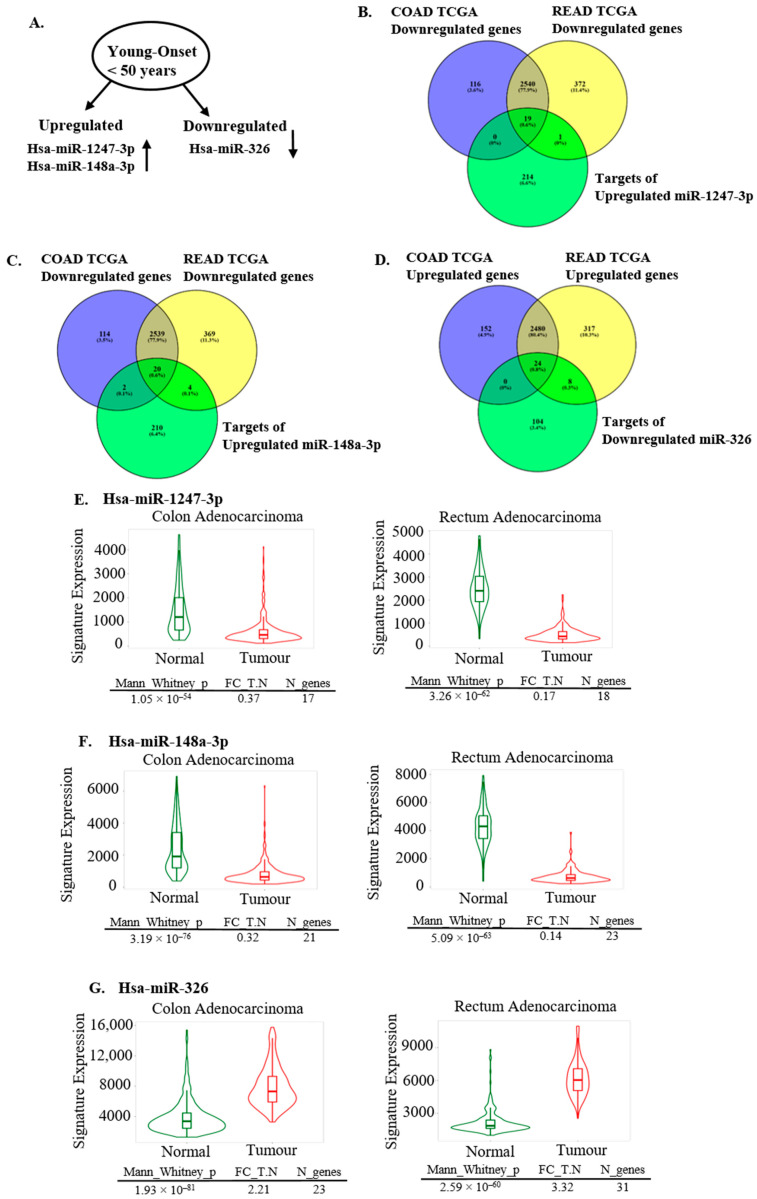
Analysis of upregulated and downregulated DEM target genes in colon and rectal cancer development and tumor onset. (**A**) Schematic representation of validated upregulated and downregulated DEMs in EOCRC. (**B**) Venn diagram comparison of downregulated genes in TCGA datasets of colon (COAD) and rectum (READ) adenocarcinoma with experimentally verified target genes of upregulated hsa-miR-1247-3p. (**C**) Venn diagram comparison of downregulated genes in TCGA datasets of colon (COAD) and rectum (READ) adenocarcinoma with experimentally verified targets of upregulated hsa-miR-148a-3p. (**D**) Venn diagram comparison of upregulated genes in TCGA datasets of colon (COAD) and rectum (READ) adenocarcinoma with experimentally verified targets of downregulated hsa-miR-326. (E and F) TNM plot distribution of the predicted COAD and READ downregulated DEM target genes of hsa-miR-1247-3p (**E**) and hsa-miR-148a-3p (**F**) between normal and tumor colon and rectum adenocarcinoma datasets using gene-chip data (https://tnmplot.com/analysis, accessed on 26 December 2024). Significant downregulation of gene expression was observed between normal and tumor tissues, indicating the importance of the predicted downregulation in tumor development. (**G**) TNM plot distribution of the predicted COAD and READ upregulated DEM target genes of hsa-miR-326 in tumor and normal tissues of colon and rectum adenocarcinoma datasets using gene-chip data (https://tnmplot.com/analysis, accessed on 26 December 2024). Expression of target genes was significantly upregulated between normal and tumor tissues, indicating importance of the upregulation in tumor development and disease onset.

**Figure 8 ncrna-11-00010-f008:**
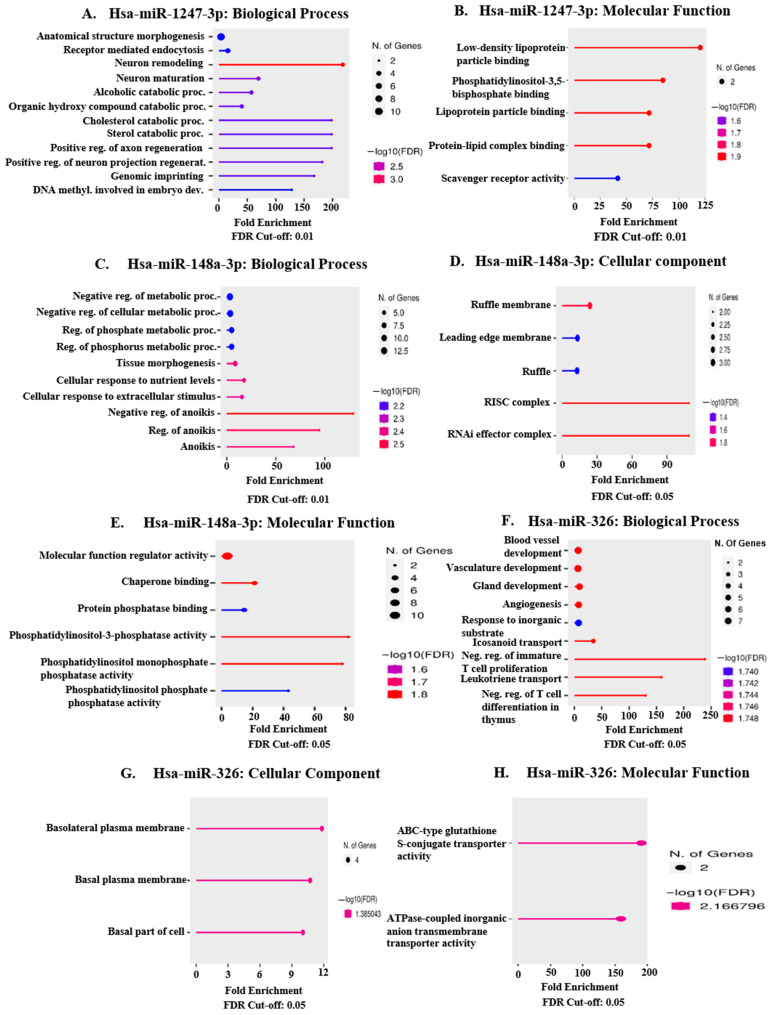
Enrichment analysis of proteins common between targets of validated miRNAs and targets differentially expressed in TCGA-COAD and TCGA-READ databases (upregulated and downregulated reciprocally to each other) using the tool ShinyGO 0.77 (sdstate.edu). (**A**) represents the gene ontology enrichment chart for the annotation Biological Process (BP) for COAD and READ downregulated targets of upregulated hsa-miR-1247-3p. (**B**) represents the enrichment chart for Molecular Function (MF) for targets of upregulated hsa-miR-1247-3p. (**C**) represents the gene ontology enrichment chart for the annotation BP for hsa-miR-148a-3p targets. (**D**) represents that for the annotation of Cellular Component (CC) for targets of hsa-miR-148a-3p. (**E**) represents the enrichment analysis for the category of MF for miR-148a-3p. (**F**) represents the BP enrichment for targets of validated downregulated miRNA hsa-miR-326. (**G**) represents the annotation chart for the CC category of hsa-miR-326 targets, and (**H**) represents enrichment for the MF category of the targets of hsa-miR-326. Genes selected by FDR and sorted by the number of pathway genes. The size of the solid circle indicates the number of miRNA targets enriched for each pathway. FDR Cut-off was taken as 0.01 for (**A**–**C**) and 0.05 for (**D**–**H**) against humans as a selected species.

**Table 1 ncrna-11-00010-t001:** Clinicopathological features of clinical samples (tumor and paired normal from each patient) sent for NGS-based miRNA Sequencing (discovery cohort).

Young/Old	Patient Number	Age	Sex	Stage	Grade	Histopathological Type	Location
Young	1	39	Female	T_3_N_2b_M_x_	G2	Adenocarcinoma	Junction of ascending and transverse colon
Young	2	44	Female	T_4a_N_0_	G2	Adenocarcinoma	Rectosigmoid colon
Young	3	47	Male	T_2_N_0_	G2	Adenocarcinoma	Ascending colon and cecum
Old	4	66	Male	T_3_N_2b_	G2	Adenocarcinoma	Rectum
Old	5	60	Female	T_1_N_0_	G1	Adenocarcinoma	Rectum

**Table 2 ncrna-11-00010-t002:** List of significantly differentially expressed miRNAs identified in our bioinformatics analysis of miRNA-sequencing in young patient tumors (EOCRC) compared with corresponding normal (DEM cut-off: *p*-value < 0.05, Log_2_ fold change: >2, <−2).

miRNA	logFC	AveExpr	*p*-Value	Upregulated/Down Regulated
hsa-miR-1247-3p	3.70881538	2.13482567	0.00174089	Up
hsa-miR-27a-5p	2.61304708	7.28367881	0.00901648	Up
hsa-miR-96-5p	2.80991462	6.01862554	0.02211184	Up
hsa-miR-148a-3p	2.49604406	1.99822506	0.02365696	Up
hsa-miR-135b-5p	3.41150654	6.19388937	0.02799316	Up
hsa-miR-133b	−2.01434615	1.8466954	0.00292072	Down
hsa-miR-133a-5p	−2.13259517	0.22441615	0.00324305	Down
hsa-miR-378e	−2.61154969	1.13114207	0.00366325	Down
hsa-miR-139-3p	−2.1809868	1.433196	0.00568353	Down
hsa-miR-378a-5p	−2.41712807	3.98406158	0.00620528	Down
hsa-miR-887-3p	−2.10734608	0.13910073	0.00627189	Down
hsa-miR-30a-5p	−2.0469516	9.41801359	0.00769662	Down
hsa-miR-9-3p	−2.15316894	1.64216281	0.00770244	Down
hsa-miR-490-3p	−2.11550509	1.30918108	0.01130573	Down
hsa-miR-143-3p	−2.29934116	15.5726929	0.01331962	Down
hsa-miR-378c	−2.49476989	7.31380586	0.01533526	Down
hsa-miR-490-5p	−2.14003278	0.29238366	0.01709929	Down
hsa-miR-326	−2.63728355	1.68062835	0.01712207	Down
hsa-miR-504-5p	−2.28494574	3.64916906	0.01904966	Down
hsa-miR-378d	−2.48909916	6.12055306	0.01914212	Down
hsa-miR-143-5p	−2.0070137	7.68776446	0.02078392	Down
hsa-miR-139-5p	−2.27780576	6.20200982	0.0240824	Down
hsa-miR-363-3p	−2.40777269	6.41529968	0.04421748	Down

**Table 3 ncrna-11-00010-t003:** List of significantly differentially expressed miRNAs identified in our bioinformatics analysis of miRNA-sequencing in old patient tumors (LOCRC) compared with corresponding normal (DEM cut-off: *p*-value < 0.05, Log_2_ fold change: >2, <−2).

miRNA	logFC	AveExpr	*p*-Value	Upregulated/Down Regulated
hsa-miR-455-3p	2.01483885	4.787073265	0.008717	Up↑
hsa-miR-31-3p	3.06621503	0.091078151	0.01037511	Up↑
hsa-miR-204-5p	−4.35495512	2.405411699	0.01160546	Down↓
hsa-miR-1247-5p	2.63591607	1.784235588	0.02448825	Up↑
hsa-miR-10524-5p	2.07155472	−0.206629246	0.02805908	Up↑
hsa-miR-549a-5p	3.15057457	0.773732542	0.03220999	Up↑
hsa-miR-934	2.11438475	−0.379786505	0.03480386	Up↑
hsa-miR-135b-3p	2.6736609	1.146870727	0.03507436	Up↑
hsa-miR-10396b-3p	2.00250344	−0.581985831	0.03772035	Up↑
hsa-miR-4485-3p	2.92834074	2.12684681	0.04423556	Up↑
hsa-miR-31-5p	4.862277188	3.243605845	0.04948822	Up↑

**Table 4 ncrna-11-00010-t004:** Clinicopathological features of clinical samples (tumor and paired normal from each patient) used for validation of NGS results.

Young/Aged	Patient Number	Age	Sex	Stage	Grade	Histopathological Type	Location
Young	Y1	27	Male	cT_4a_N_2_M_0_	G2	Adenocarcinoma	Ascending colon
Young	Y2	25	Male	cT_4_N_2a_	G3	Adenocarcinoma	Lower Rectum
Young	Y3	42	Male	cT_3_N_0_M_0_	G2	Adenocarcinoma	Transverse colon
Young	Y4	39	Male	cT_3_N_2_	G1	Adenocarcinoma	Transverse colon
Young	Y5	11	Male	cT_3_N_1_M_0_	G3	Adenocarcinoma	Transverse colon
Young	Y6	18	Female	T_4b_N_2_M_1b_	G3	Adenocarcinoma	Upper rectum
Young	Y7	32	Male	cT_3_N_0_M_0_	G2	Adenocarcinoma	Sigmoid colon
Young	Y8	45	Female	cT_4_N_0_M_1c_	G1	Adenocarcinoma	Upper rectum
Young	Y9	18	Female	cT_3_N_2_M_0_	G2	Adenocarcinoma	Lower rectum
Young	Y10	43	Male	cT_4a_N	G2	Adenocarcinoma	Distal rectum
Young	Y11	37	Female	pT3N0	G2	Adenocarcinoma	Rectosigmoid
Young	Y12	49	Female	pT3N2a	G2	Adenocarcinoma	Caecum
Young	Y13	47	Female	pT_3_N_0_	G2	Adenocarcinoma	Sigmoid colon
Young	Y14	35	Male	pT_3_N_2_M_0_	G2	Adenocarcinoma	Caecum
Young	Y15	48	Male	pT_3_N_0_M_0_	G2	Adenocarcinoma	Sigmoid colon
Young	Y16	26	Male	pT_3_N_0_M_0_	G1	Adenocarcinoma	Transverse colon
Aged	O1	58	Male	cT_3_N_2_M_0_	G2	Adenocarcinoma	Ascending colon
Aged	O2	70	Male	T_4b_N_2_M_1b_	G3	Adenocarcinoma	Lower rectum
Aged	O3	61	Male	cT_3_N_2_M_0_	G2	Adenocarcinoma	Ascending colon
Aged	O4	75	Male	T_3_N_2_M_0_	G3	Adenocarcinoma	Hepatic Flexure
Aged	O5	59	Male	T_2_N_0_M_0_	G2	Adenocarcinoma	Lower rectum
Aged	O6	63	Male	T_4_N_2b_M_1a_	G2	Adenocarcinoma	Rectosigmoid,
Aged	O7	57	Female	cT_3_N_1_M_0_	G2	Adenocarcinoma	Hepatic Flexure
Aged	O8	59	Female	pT3N0	G2	Adenocarcinoma	Rectum
Aged	O9	59	Female	cT_4_N_0_	G2	Adenocarcinoma	Sigmoid colon
Aged	O10	62	Female	cT_3_N_2_	G2	Adenocarcinoma	Ascending colon
Aged	O11	53	Male	cT_4_N_1_	G2	Adenocarcinoma	Lower rectum

**Table 5 ncrna-11-00010-t005:** List of primary antibodies used.

Antigen Name	Raised in	Source	Dilution for Immunohistochemistry	Catalog No.
MLH1	Mouse Monoclonal	PathnSitu	1:100	CM098
MSH2	Rabbit monoclonal	PathnSitu	1:100	CR055
MSH6	Rabbit monoclonal	PathnSitu	1:100	CR056
PMS2	Rabbit monoclonal	PathnSitu	1:50	CR067
Beta-catenin	Mouse monoclonal	Thermo Fisher Scientific, Waltham, MA, USA	1:100	13–8400

**Table 6 ncrna-11-00010-t006:** List of PCR primers.

Sl No	Name	Sequence (5′–3′)	Annealing Temperature
1	APC1 Forward	GAGGCAGAATCAGCTCCATCCAAG	60 °C
2	APC1 Reverse	CTTCTGCTTGGTGGCATGGTTTGTC	60 °C
3	APC 2 Forward	GCAGACTGCAGGGTTCTAGTT	57 °C
4	APC 2 Reverse	AGAGCACTCAGGCTGGATGA	57 °C
5	APC 3 Forward	GACAAACCATGCCACCAAGCAGAAG	60 °C
6	APC3 Reverse	CACAATACACCCGTGGCATATCATC	60 °C

**Table 7 ncrna-11-00010-t007:** Primer sequences for Reverse Transcription (RT) Stem Loop Primers (SLP).

miRNA	RT USLP Sequence (5′–3′)
miR-135b-5p	GAAAGAAGGCGAGGAGCAGATCGAGGAAGAAGACGGAAGAATGTGCGTCTCGCCTTCTTTCTCACATAG
miR-148a-3p	GAAAGAAGGCGAGGAGCAGATCGAGGAAGAAGACGGAAGAATGTGCGTCTCGCCTTCTTTCACAAAGTT
miR-1247-3p	GAAAGAAGGCGAGGAGCAGATCGAGGAAGAAGACGGAAGAATGTGCGTCTCGCCTTCTTTCGCTCCAGT
miR-326	GAAAGAAGGCGAGGAGCAGATCGAGGAAGAAGACGGAAGAATGTGCGTCTCGCCTTCTTTCCTGGAGGA
miR-27a-5p	GAAAGAAGGCGAGGAGCAGATCGAGGAAGAAGACGGAAGAATGTGCGTCTCGCCTTCTTTCTGCTCACA
miR-96-5p	GAAAGAAGGCGAGGAGCAGATCGAGGAAGAAGACGGAAGAATGTGCGTCTCGCCTTCTTTCAGCAAAAA
miR-378a-5p	GAAAGAAGGCGAGGAGCAGATCGAGGAAGAAGACGGAAGAATGTGCGTCTCGCCTTCTTTCACACAGGA
miR-378c	GAAAGAAGGCGAGGAGCAGATCGAGGAAGAAGACGGAAGAATGTGCGTCTCGCCTTCTTTCCCACTCTT
miR-378d	GAAAGAAGGCGAGGAGCAGATCGAGGAAGAAGACGGAAGAATGTGCGTCTCGCCTTCTTTCTTTCTGAC
U6 snRNA	CGCTTCACGAATTTGCGTGTCA

**Table 8 ncrna-11-00010-t008:** Primer sequences for Real-time PCR forward and reverse primers.

**miRNA**	**Forward Primer**	**Tm (°C)**
miR-135b-5p	GCTATGGCTTTTCATTCCTATGTGA	58
miR-148a-3p	TCAGTGCACTACAGAACTTTGT	57
miR-1247-3p	AACGTCGAGACTGGAGC	59
miR-326	GCCCTTCCTCCAGGAAA	59
miR-27a-5p	GGGCTTAGCTGCTTGTGA	59
miR-96-5p	TTTGGCACTAGCACATTTTTGCT	59
miR-378a-5p	GACTCCAGGTCCTGTGT	58
miR-378c	GACTTGGAGTCAGAAGAGTGG	58
miR-378d	CACTGGACTTGGAGTCAGAAA	57
U6 snRNA	GCTTCGGCAGCACATATACTAAAAT	58
	**Reverse primer**	**Tm (°C)**
miRNAs	CGAGGAAGAAGACGGAAGAAT	57
U6 reverse	CGCTTCACGAATTTGCGTGTCAT	62

## Data Availability

The data discussed in this publication are deposited in NCBI’s Gene Expression Omnibus [81] and are openly accessible through GEO Series accession number GSE236991 (https://www.ncbi.nlm.nih.gov/geo/query/acc.cgi?acc=GSE236991, accessed on 29 January 2025). Validation data supporting the findings of this study are available from the corresponding authors upon request due to ethical reasons.

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
