# Peer review of "Differential Expression of miRNAs Between Young-Onset and Late-Onset Indian Colorectal Carcinoma Patients"

_ncrna, 2025, doi:10.3390/ncrna11010010_

Round 1

Reviewer 1 Report (Previous Reviewer 1)

Comments and Suggestions for Authors

attached

Author Response

Reviewer 2 Report (Previous Reviewer 2)

Comments and Suggestions for Authors

I am ok with the current version. The authors have addressed my point in the revision.

Author Response

Reviewer 3 Report (New Reviewer)

Comments and Suggestions for Authors

he submitted manuscript presents the results of a study investigating differentially expressed microRNAs (miRs) in two groups of patients: those with early-onset colorectal cancer (CRC) (below 50 years old) and those with late-onset CRC. The authors aimed to identify miRs, predict their putative targets in CRC, and analyze enriched gene sets associated with early-onset CRC using two patient cohorts from Indian hospitals and CRC patients from the TCGA-COAD dataset. The study provides interesting results that are properly interpreted, well-discussed, and supportive of the final conclusions. However, the manuscript has several limitations that must be addressed before the study can be considered for acceptance.

The discussion, particularly its initial paragraphs, should be more concise and specific, avoiding a recapitulation of the results and repetition of facts already presented in the Introduction.

The authors should explain why they used only the COAD cohort to validate their CRC results, despite the TCGA also providing the READ cohort. Merging the READ and COAD cohorts might provide a better validation cohort, as the patients from the Indian cohorts included both colon and rectal lesions. If tumor localization in the rectum was an exclusion criterion, this should be clearly stated in the Methods section, specifically in the patient recruitment paragraph.

The enrichment analysis presented in Figure 8 should be repeated using less general levels of gene ontology. Including more detailed Biological Processes (BPs), Molecular Functions (MFs), and Cellular Components (CCs) could provide much more informative insights.

The manuscript contains numerous typographical errors, inconsistencies in spacing, and issues with the use of brackets, units, and formatting. These should be thoroughly corrected in the revised version of the manuscript to ensure clarity and consistency.

The descriptions of experiments, particularly the IHC and PCR methods, require significant improvements. More detailed information should be provided to ensure that the methods can be accurately reproduced if necessary.

Vendor details should include detailed information, such as the city, state/province (if applicable), and country, when mentioned for the first time. Afterward, only the vendor's name should be used.

The inline citations of webpages and documents should be revised to follow the editor's guidelines.

line 54-56 - difficult to understand, please revise
line 43 - "all" or "males only"?

Author Response

Reviewer 4 Report (New Reviewer)

Comments and Suggestions for Authors

INTRODUCTION-->The introduction is too broad, it would be important to focus only on the fundamental informations.  Put a brief but clear phrase that describe the aim of your study. It's not necessary to explain into the introduction the draw of the study (from phrase 159 to 172).

CONCLUSIONS--> Also in this paragraph it is important to focus on the very important informations. From the line 568 to 580 you described again the draw of you study.It's important to understund what you find. Wich are the miRNAs resulted statistically significant? Wich are the target genes identified?

Author Response

Reviewer 5 Report (New Reviewer)

Comments and Suggestions for Authors

1. Improving the English language in text.

2. In the Results section entitled "2.1. Patient recruitment and sample collection" please specify how many samples correspond to EOCRC and how many to LOCRC.  

3. You should also show in a table the 11 differentially expressed miRNAs you found in the LOCRC group

4. You could also perform TCGA analysis for the TCGA-READ dataset.  Why didn't you perform the TCGA analysis using the 11 differentially expressed miRNAs you found in the LOCRC group?

5. It should be highlighted that contrary to expectation, the age-specific miRNA expression analysis in the TCGA-COAD dataset (distinguishing the population into four subsets - Young Normal, Old Normal, Young Tumour and Old Tumour) detected a slight upregulation even in the previously identified downregulated miRNAs (hsa-miR-326,hsa-miR-378a-5p)

6. Why did you choose for validation on samples the miRNAs hsa-miR-1247-3p, hsa-miR-27a-5p, hsa-miR-96-5p, hsa-miR-326, hsa-miR-378a-5p hsa-miR148a-3p hsa- miR-135b-5p, hsa-miR-378c and hsa-miR-378d if the miRNAs hsa-miR-326 and hsa-miR-378a-5p provided discordant results from the previous TCGA analysis?

7. In materials and methods, in the section ‘Patient recruitment-Biospecimen collection’ specify which cohorts are used for the study and how many patients/samples they consist of.

8. In materials and methods, move Information on PCR primers, from statistical analysis to miRNA validation by qRT-PCR.

Round 2

Reviewer 3 Report (New Reviewer)

Comments and Suggestions for Authors

The authors have properly addressed the reviewer's remarks, and the revised version of the manuscript demonstrates improved quality in the study design (including the addition of READ and gene ontology/enrichment) and the presentation of results. The manuscript could be considered for acceptance if the following (minor) remarks are considered by the authors:

The TCGA-COAD and TCGA-READ datasets should have been merged into a single validation dataset; however, the authors have added TCGA-READ only as an independent cohort. However, this issue does not impact the overall quality of the study.

The manuscript requires thorough proofreading and correction of minor spelling mistakes/typos, inconsistencies in the use of capital and lowercase letters (e.g., for words in brackets), and spaces before/after mathematical operators (e.g., <, >, =).

Lines 30, 144, 184, 810 (and wherever else applicable): TCGA-COAD and READ should be written as TCGA-COAD and TCGA-READ.

Vendor affiliations in the Methods section: Use the full affiliation the first time only; subsequent references should include the vendor's name only.

Author Response

Reviewer 5 Report (New Reviewer)

Comments and Suggestions for Authors

Dear Authors,

all of my previous comments were properly addressed. The manuscript was significantly improved and can be accepted for publication after the editorial check.

Author Response

This manuscript is a resubmission of an earlier submission. The following is a list of the peer review reports and author responses from that submission.

Round 1

Reviewer 1 Report

Comments and Suggestions for Authors

Reviewer 2 Report

Comments and Suggestions for Authors

Summary:

In this manuscript, the authors investigate the differences in miRNA expression between young and old colorectal cancer (CRC) patients. The study comprises several key steps: miRNA-seq analysis from five different patient samples, validation using TCGA independent cohort, further validation with qPCR from additional patient samples, and prediction of related miRNA targets with functional analysis using GO terms.

Major Concerns:

Age-Specific Basal miRNA Levels:

The comparison of miRNA responses between young and old patients is based on tumor vs. normal tissue fold changes or differences in tumors. It is essential to consider the basal miRNA levels in normal tissues for both age groups. The authors should analyze and present the miRNA levels in normal tissues from young and old patients to ensure accurate comparisons.

TCGA Data Analysis (Figure 2):

The TCGA data analysis in Figure 2 does not incorporate age-specific comparisons.The TCGA data should be divided into four groups: young-normal, young-tumor, old-normal, and old-tumor. The authors should plot these four groups to clearly illustrate the age-related differences in miRNA expression.

Emphasis on Age Differences:

Although the title and mainly focus of the paper are on miRNA differences between young and old CRC patients, most of the figures only show tumor vs. normal tissue comparisons. This does not provide sufficient evidence or information regarding age-specific differences.

Expand Sample Size: While the initial analysis involved only five patients, and Fig 4-5 only include 3 samples, expanding the sample size would increase the robustness and generalizability of the findings.

Minor point: In Figure 6, the miRNA validation is presented using delta Ct values. Instead of delta Ct values, the authors should present normalized expression levels relative to U6 snRNA. 

Comments on the Quality of English Language

none